# Predicting natural language descriptions of mono-molecular odorants

E. Darío Gutiérrez[1], Amit Dhurandhar[2], Andreas Keller[3], Pablo Meyer [1,4] & Guillermo A. Cecchi [1]

There has been recent progress in predicting whether common verbal descriptors such as "fishy", "floral" or "fruity" apply to the smell of odorous molecules. However, accurate predictions have been achieved only for a small number of descriptors. Here, we show that applying natural-language semantic representations on a small set of general olfactory perceptual descriptors allows for the accurate inference of perceptual ratings for mono-molecular odorants over a large and potentially arbitrary set of descriptors. This is noteworthy given that the prevailing view is that humans' capacity to identify or characterize odors by name is poor. We successfully apply our semantics-based approach to predict perceptual ratings with an accuracy higher than 0.5 for up to 70 olfactory perceptual descriptors, a ten-fold increase in the number of descriptors from previous attempts. These results imply that the semantic distance between descriptors defines the equivalent of an odorwheel.

[1] Computational Biology Center, T.J. Watson IBM Research Laboratory, 1101 Kitchawan Rd, Yorktown Heights, NY 10598, USA. [2] Artificial Intelligence Foundations, T.J. Watson IBM Research Laboratory, 1101 Kitchawan Rd, Yorktown Heights, NY 10598, USA. [3] AK Consulting, 508 East 78th Street, Apt 5N, New York, NY 10075, USA. [4] Department of Genetics and Genomic Sciences, Icahn School of Medicine at Mount Sinai, New York, NY 10029, USA. Correspondence and requests for materials should be addressed to P.M. (email: pmeyerr@us.ibm.com) or to G.A.C. (email: gcecchi@us.ibm.com)

Humans are unique in their capacity to express sensory perceptual experiences using the powerful machinery of language—e.g. "The *Requiem* is Mozart's most *mournful* work", "I feel a *stabbing* more than a *burning* pain", which prompts the question: to what extent can language adequately convey perception? This question is particularly contentious in the realm of olfaction research, where quantifying odor percepts through semantic attributes is a central endeavor[1,2]. Indeed ratings of a molecule along a comprehensive set of descriptors such as "putrid", "floral", and "apple" could uniquely characterize the molecule's odor[1], and experts spend considerable time and effort handcrafting domain-specific sets of odor descriptors or collecting ratings for large numbers of descriptors for each molecule of interest[3,4]. A standard, generally applicable set of "primary" odor descriptors would be more amenable[5] but despite decades of research this effort has been in vain[2,6]. The prevailing view is that there is a significant disconnect between humans' strong capacity for odor discrimination[7] and their inability to identify or characterize odors by name[1,8–12]. This would seem to suggest that semantic descriptors cannot be reliable. Yet semantically generated multidimensional descriptors have been proven to be stable[13] and there is substantial evidence of interactions between language and various perceptual modalities including olfaction[14–19]. Recent work even suggests that olfactory knowledge can improve the performance of linguistic representations in predicting human similarity judgments[20], while linguistic representations can be applied to quantify the olfactory specificity and familiarity of words[6]. Most recent work uses the linguistic approach to predict a reduced representation, via clustering, of the odor of a molecule[21]; however, the predictive efficacy of this model falls abruptly when more than 5 clusters of descriptors are considered.

We here show that applying natural-language semantic representations on a small set of general olfactory-perceptual descriptors can allow for the accurate inference of perceptual ratings for mono-molecular odorants over a large and potentially arbitrary set of descriptors. Furthermore, combining such semantic-based perceptual ratings predictions with a molecule-to-ratings model that relies on chemoinformatic features, we perform zero-shot learning inference[20,22] of perceptual ratings for arbitrary molecules.

## Results

**Correspondence between semantic space and olfactory ratings space.** To investigate whether semantic representations derived from language use could be applied to reliably predict how molecules are rated along a large set of detailed olfactory-perceptual descriptors, we chose to predict the ratings of 146 fine-grained odor descriptors of the well known Dravnieks dataset (Fig. 1)[23]. The ratings are obtained by asking human raters to assign values, on a fixed scale, of how close their perceptual experience of smelling an odorant is to each one of the descriptors (see Methods). As a starting point to learn a generalizable semantic-perceptual model, we used the ratings from the 19 general descriptors of a different study, the DREAM dataset[24] as it has 58 molecules in common, from 128 in total, and shares 10 descriptors with the Dravnieks dataset (Fig. 1 and Supplementary Table 1). To quantify the semantic relationship between the DREAM and Dravnieks descriptors, we used a representation of linguistic data known as distributional semantic models. These models are quantitative, data-driven, vectorial representations of word meaning motivated by the distributional hypothesis, which asserts that the meaning of a word can be inferred as a function of the linguistic contexts in which it occurs[25]. A distributional semantic model assigns a vector to each word in a lexicon, based on the word's use in language; words that are used in similar contexts, thus assumed to be more semantically similar, have vectors that are closer together in the distributional semantic space of the model (Fig. 1b). In particular, we utilized publicly available 300-dimensional semantic vectors produced using the fastText skip-gram algorithm that were trained on a corpus of over 16 billion words[26]. The fastText model contained vectors corresponding to the 19 DREAM descriptors which we refer to as the *DREAM semantic vectors*, and to 131 of the 146 Dravnieks

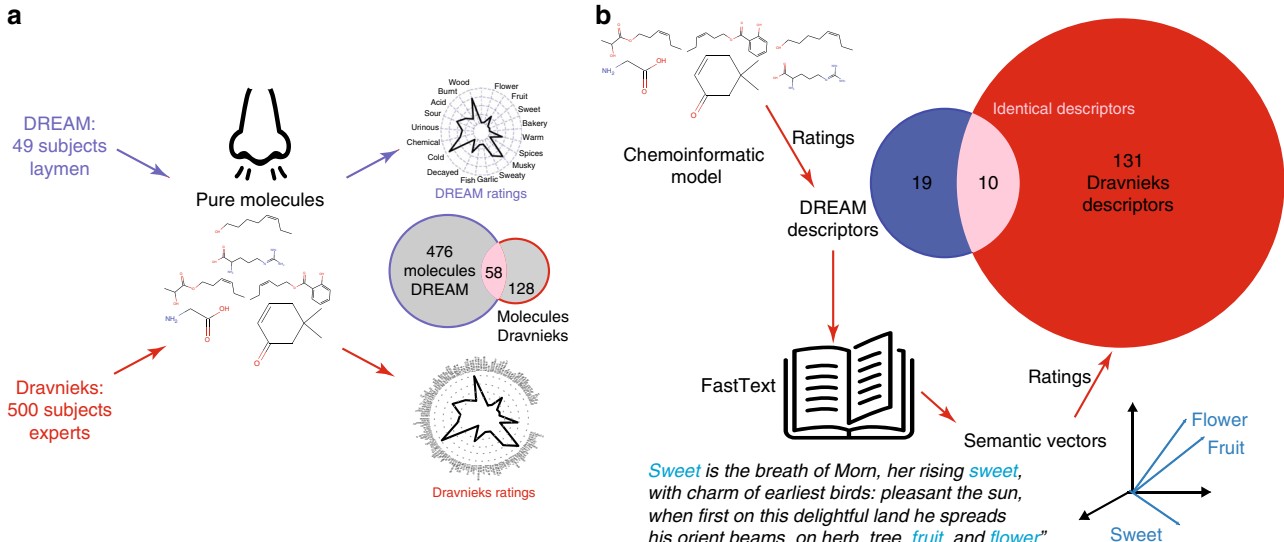

**Fig. 1** Construction of a universal perceptual map. **a** Diagram describing similitudes and differences between the DREAM and Dravnieks psychophysical olfactory datasets. Dravnieks dataset used olfaction experts, a smaller set of 128 molecules, 58 overlapping with DREAM, and a large set of 146 descriptors, 10 descriptors overlapping with DREAM. **b** Diagram showing the approach to predict ratings for the Dravnieks descriptor sets. For non-overlap molecules between the datasets, a chemoinformatic model helps predicting ratings for values of the DREAM set of 19 perceptual descriptors. We then use fastText to generate semantic vectors for the DREAM and Dravnieks descriptors by searching for co-occurrence of words in sentences as shown in the example (a fragment of Milton's *Paradise Lost*). A model using these vectors is then applied to DREAM ratings/predictions to generate Dravnieks rating values for 131 descriptors

descriptors which we refer to as the *Dravnieks semantic vectors*. Note that the training corpus was not biased in any way to include more or less olfaction- or perception-related material, i.e., it was intended to represent the general structure of semantic knowledge.

Given that the DREAM and Dravnieks studies presented different sets of descriptors to the subjects, we expect that the perceptual ratings for the molecules in common will be re-anchored according to the available descriptors, and consequently that the descriptor ratings for the two datasets will differ even on shared descriptors[27]. Indeed we find that, although the correlations across the 58 shared molecules are high for the 19 corresponding descriptors (Fig. 2a), the highest correlation is not always between the matching descriptors: e.g., although "sweet" in DREAM is most highly correlated to "sweet" in Dravnieks, "fruit" has a higher correlation to "peach" than to "fruity" (Supplementary Table 1). Nonetheless, the clusters of highly correlated descriptors defined by the dendrogram follow the close semantic relationship between the descriptors—e.g., "flower" from DREAM correlates highly with the co-clustered "rose", "violets", "incense", "perfumery", "cologne", "floral", and "lavender" from Dravnieks.

We compared the correlation matrix based on the descriptors' perceptual ratings (Fig. 2a) to a correlation matrix between the DREAM and Dravnieks semantic vectors (Fig. 2b). We observe that the two correlation matrices are similarly structured (Procrustes dissimilarity $p < 0.05$ tested against randomized surrogates, correlation between maxima across the DREAM descriptors is $r = 0.74$, $p < 10^{-4}$ and $r = 0.5$, $p < 10^{-9}$ across Dravnieks descriptors). This is also reflected in the semantic vector correlation matrix where "sweet" is similarly maximally correlated with "sweet" in Dravnieks and "fruit" correlation is with "peach" and "citrus" than with "fruity". Finally although "flower" shares a large weight with "floral", it has similar correlation with "strawberry", "fragrant", and "lavender" (Fig. 2b, top). Further insight is gained from looking at arrangement changes of two-dimensional projections of the DREAM descriptors based on their ratings distance (Fig. 2c) and their semantic distance (Fig. 2d; also see Methods). Notably, we observe only small local distortions of group mappings, e.g., "grass", "flower" and "fruit" contiguous in both spaces (pink). However, there is also a global distortion as "sweet" is arranged in the semantic space near its antonym "sour," and in the ratings space "sweet" is arranged closer to the perceptually similar term "bakery," and "sour" is arranged closer to the perceptually similar term "decayed".

**Extending predictions to arbitrary descriptors**. The similarities in how the descriptors are arranged in the olfactory-perceptual space and in the semantic space favor the hypothesis of a tight perceptual-linguistic bond between the descriptors ratings and their linguistic meanings. Consequently we developed a model that learns a transformation **S** from the 19 DREAM semantic vectors to the 131 Dravnieks semantic vectors (Fig. 3a, top left) and refer to this model as the direct semantic model. We hypothesized that, given the correspondences between the perceptual and semantic spaces, we could use this same matrix **S** to predict the ratings of the 131 Dravnieks descriptors based solely on the ratings of the 19 DREAM descriptors and the semantic relation between the DREAM and Dravnieks descriptors. We compared the results of the semantic model to a direct ratings model that uses a training set of molecules for which both DREAM and Dravnieks ratings are available to learn a transformation **R** that can predict a new molecule's ratings on the Dravnieks descriptors, given its ratings on the DREAM descriptors (Fig. 3a, top

right). To further investigate the complementarity of the information provided by the semantic vectors and ratings data, we also looked at the performance of a mixed model that averaged the predictions of the two models.

To avoid overfitting, we used a cross-validation procedure where the 58 shared molecules are repeatedly divided at random into test sets and training sets and results averaged over repetitions. The performance of all three models was evaluated as the number of training molecules is varied. We compared each model's performance by computing the median of the correlation between the predicted ratings and the actual ratings for a test set of molecules, across the Dravnieks descriptors. As ratings of molecules across descriptors are significantly correlated, we defined as an appropriate baseline prediction the mean rating for each descriptor across all molecules used for training the model and found that this baseline correlation is around 0.6. We then calculate a $Z$-score that compares the difference between the baseline correlation and the correlations produced by the models, taking into account their dependence. We report the median $Z$-score across molecules and across repetitions of cross-validation.

Remarkably, without making use of any of the ratings from the target set, i.e., an instance of zero-shot learning[22], the semantic model is able to predict the ratings in the target set reasonably well (Fig. 3a bottom and 3b inset) with a median $Z = 3.7$, $r = 0.47$, $p < 10^{-4}$ (one-sided $t$-test, see Supplementary Fig. 1 for correlations plot) and better than the ratings model when trained on fewer than 6 overlapping molecules (Fig. 3a bottom, blue and gold lines). Furthermore, the mixed model showed excellent performance with a $Z$-score of up to 5 and was never outperformed by the ratings model, underscoring the importance of the contribution from the semantic model and suggesting complementarity between information available in the ratings and the semantic model (Fig. 3a bottom, green and gold lines).

**Extending predictions to arbitrary molecules**. We extended this approach using a chemoinformatics-to-perception model that allows the prediction of ratings along the 19 DREAM descriptors for any molecule using its molecular features[24]. We used an imputation model **C**, pre-trained with the DREAM dataset, to predict the 70 Dravnieks molecules that are not part of the DREAM dataset (Fig. 3b, top row; see also Methods). **C** is then combined with either the semantic transformation **S** to yield the imputed semantics model or used to train **R** yielding an imputed ratings model, both inferring Dravnieks ratings (Fig. 3b, middle row). These models were also averaged to produce a mixed model and scored on Dravnieks ratings (Fig. 3b, bottom row).

Once again, predictions of descriptor ratings based on the semantic vectors alone with no molecular training data, are significantly better than chance when no training molecules are available (Fig. 3b, bottom, median $Z = 3.4$, $r = 0.40$, $p < 0.001$, see methods—see plot inset and Supplementary Fig. 2 for correlations plot) and outperform the imputed ratings model when less than 10 molecules are available for training (Fig. 3b, bottom, blue and gold lines). We also again observe that a mixed model dominates the ratings model, showcasing the utility of semantic vectors even when ratings for a training set of molecules are available (Fig. 3b, bottom, gold and green lines). This advantage persists even as the number of molecules for which the source ratings available grows larger.

**Analysis of predictive performance**. To understand the performance of the semantics-based models, we varied the number of source DREAM descriptors whose semantic vectors are available for training the direct and imputed semantic models while using

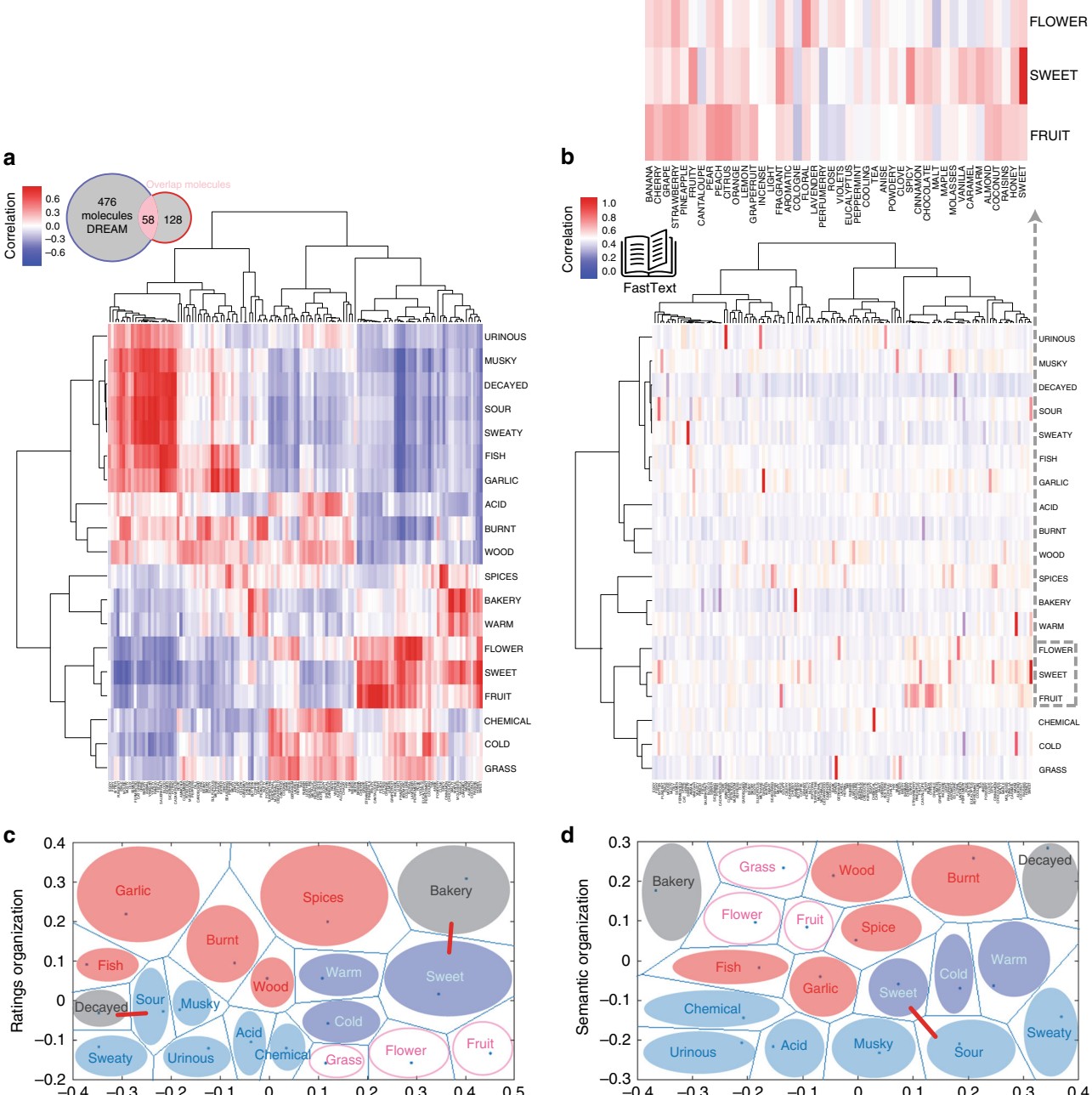

**Fig. 2** Olfactory-perceptual map and semantic structure similarities. **a** Heatmap showing correlations between the ratings of the Dravnieks descriptors (horizontal axis) and the DREAM descriptors (vertical axis) across the 58 shared molecules. Descriptors are arranged using hierarchical clustering, showing that they naturally cluster into semantically interpretable categories in the perceptual ratings correlation space. **b** Heatmap showing the correlation for the semantic vectors of the DREAM descriptors (horizontal axis) to the semantic vectors of the Dravnieks descriptors (vertical axis). Descriptors are arranged using the hierarchical clustering of **a** in order to allow direct comparison and emphasize common structure. Top: zooming in on one of the semantic clusters. Weights are not always highest between identical Dravnieks and DREAM descriptors (e.g., "flower"). **c** Organization of descriptors in the olfactory rating space through a 2D dimensionality-reduction based on the intrinsic similarity between the ratings representation of the descriptors across all molecules in the DREAM dataset. **d** Organization of descriptors in the olfactory rating space through a 2D dimensionality-reduction based on the intrinsic similarity between the semantic representation of the descriptors in the DREAM dataset. Note the locality preservation of descriptor groups, identified by colors (e.g., pink for "grass," "flower" and "fruit"), and the global distortion due to the proximity of the antonyms "sweet" and "sour" shown with a red bar in the semantic space, in turn mapped close to the perceptually closer "bakery" and "decayed," respectively, shown with a red bar in the ratings organization

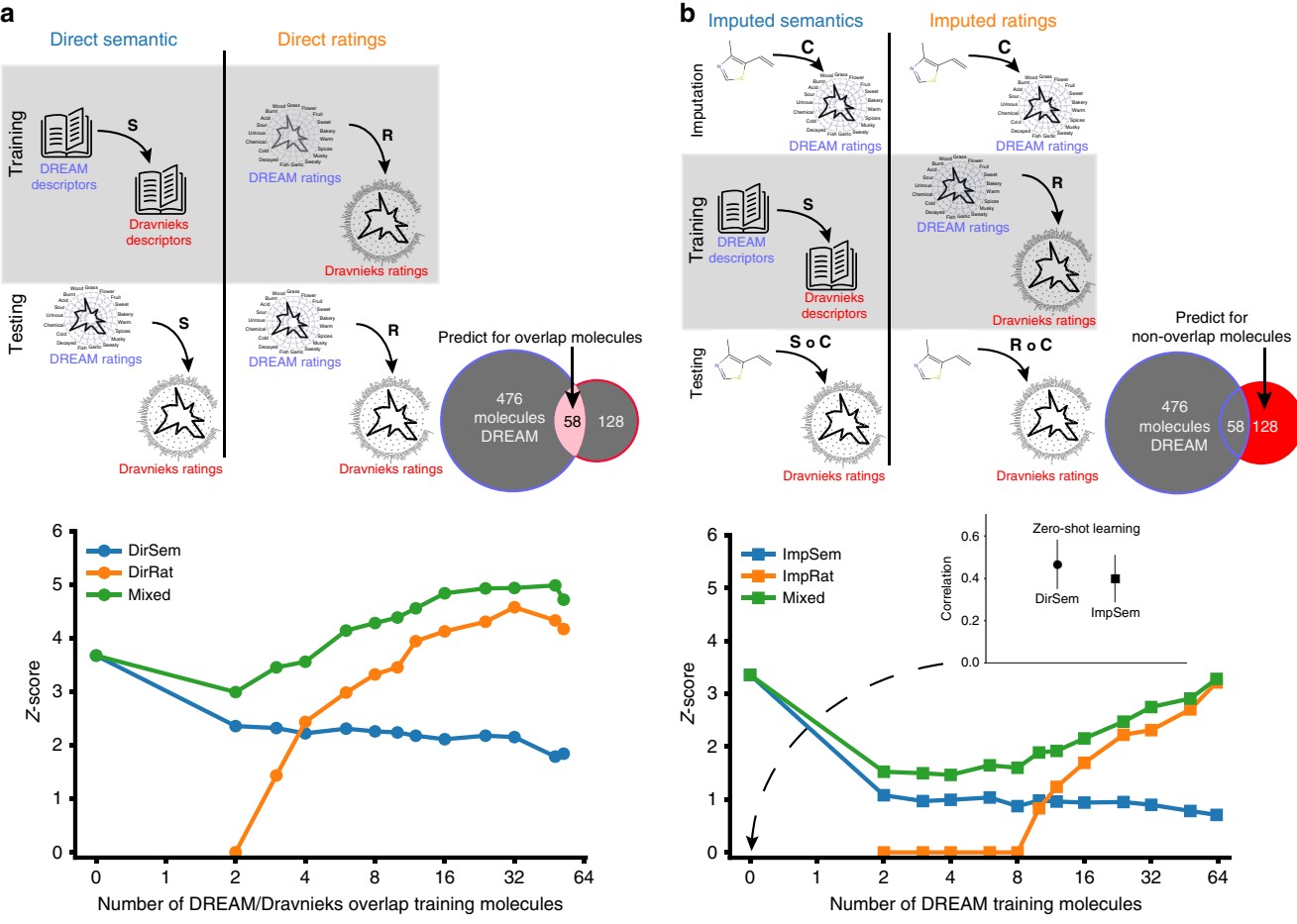

**Fig. 3** Predicting olfactory perception across descriptor sets and molecules. **a** Top Schematic of the direct models for predicting ratings. During training (top row), the direct semantic model (*DirSem* left column) learns a transformation **S** from DREAM descriptors' semantic vectors to Dravnieks descriptors' semantic vectors. Direct ratings model (*DirRat* right column) learns a transformation **R** from molecule ratings on DREAM descriptors to molecule ratings on Dravnieks descriptors. During testing (bottom row), the *DirSem* and *DirRat* models use transformations **S** and **R**, respectively, to predict molecule ratings on Dravnieks descriptors from the ratings given on DREAM descriptors. Note that during training, *DirSem* uses no molecules while *DirRat* uses the shared set of 58 molecules. Both models are tested on these 58 molecules, averaging across 100 repetitions of 10-fold cross-validation. *Bottom*: The performance of *DirSem* (blue dots) and *DirRat* (orange dots) as well as a their averaged mixed model (green dots), as the number of molecules used in training is increased. **b** Top: Schematic of the indirect models for predicting ratings. During imputation (top row), both models learn the same transformation **C** from chemoinformatic properties to the ratings on the DREAM descriptors. During training (middle row), the two models imputed semantics *ImpSem* and imputed ratings *ImpRat* learn transformations **S** and **R** using the same procedure as the training phase of *DirSem* and *DirRat*, respectively. During testing (bottom row), the *DirSem* and *DirRat* models use the transformations **S ∘ C** and **R ∘ C**, respectively, to predict molecule ratings on Dravnieks descriptors from the ones given on DREAM descriptors. Note that the *ImpSem* model uses no molecules during training, while the *ImpRat* model uses molecules from the set of 70 molecules present only in the Dravnieks dataset during training. Both models are tested on these 70 molecules, using cross-validation. Bottom: The performance of the *ImpSem* (blue squares) and *ImpRat* (orange squares) models and the mixed model (green squares), as the number of molecules used in training is increased. Inset shows the value of the correlations for the *DirSem* (black dots) *ImpSem* (black squares) when no molecules are used during training

leave-one-out cross-validation on their respective training/test molecule sets. The method we used for prioritizing the 19 perceptual descriptors is a state-of-the-art prototype selection algorithm based on a non-negative constrained reconstruction of the original data (see Methods and ref.[28]). We chose this approach as it selects recursively the best *individual* descriptor, i.e. the descriptor that best explains the entire perceptual ratings data, as opposed to commonly-used dimensionality-reduction factorization algorithms. We observe that for both models, as the number of source descriptors increases, prediction performance generally increases, though the performance improvements plateau twice at four source descriptors, notably "sour", "urinous", "burnt" and "sweet", and then around ten source descriptors (Fig. 4a). The

direct semantic model uses real DREAM ratings for making its predictions and so its correlation across descriptors is overall higher and the difference grows at the second plateau (Fig. 4a, squares and circles). This also suggests that it is possible to achieve good prediction performance on the target descriptors' predictions by collecting only a small number of ratings from a smaller number of source descriptors.

We analyzed the quality of model predictions for each of the 58 overlapping molecules of this leave-one-out model (last green dot in Fig. 3a and see Supplementary Data 1 for all the predictions) and find that the mixed model is more stable across molecules than the semantic model and as expected its correlations are also higher, around 0.8 on average (Fig. 4b).

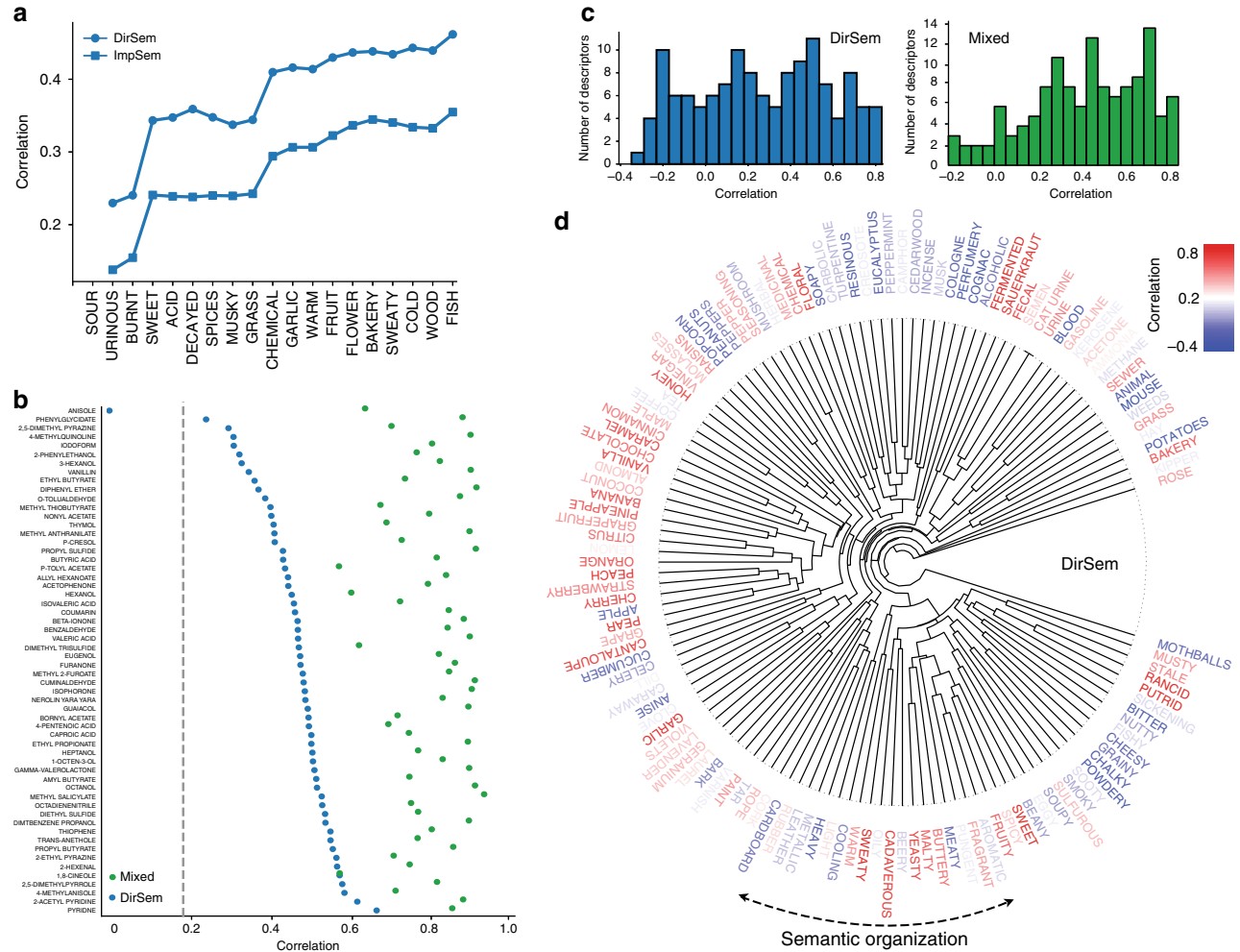

**Fig. 4** Analysis of predictive performance and map structure. **a** The performance of the direct semantic *DirSem* and the imputed semantic *ImpSem* models (open blue circles and squares, respectively) as the number of descriptors used during training is increased. **b** Prediction performance for each molecule, as measured by average correlation across descriptors between the ground truth ratings and the ratings predicted by the *DirSem* and mixed models (blue and green dots, respectively). The best-predicted molecules are toward the bottom of the chart, limit of significance (for *DirSem* model) is shown by the dotted gray line. **c** Histograms showing the median correlation across molecules for each descriptor for left the *DirSem* model and right the mixed model. **d** Odor wheel: prediction performance for each descriptor—as measured by the correlation across molecules between the ground truth and the predictions from the *DirSem* model—is indicated by the color of the text (see bar for scale). Descriptors are arranged and clustered based on their semantic vectors

Notably the semantic model predicted the perceptual ratings profile for 57 of the 58 shared molecules with significantly above-chance correlations. The best-predicted molecule *pyridine*, with a fish-like smell, had a correlation around 0.6 while the other top five predicted molecules had herbal and fruit-like smells (Fig. 4b). We also analyzed the quality of the semantic model predictions for each of the 131 Dravnieks descriptors by displaying the median correlation across molecules for each descriptor in a histogram (Fig. 4c, left). Notably about 30 percent of descriptors were predicted with a correlation higher than 0.5 for the semantic model, a value that increased to 50 percent of the descriptors for the mixed model (Fig. 4c, right).

**Organization of descriptors in semantic and olfactory ratings spaces**. A closer look at the nature of the semantic and perceptual ratings spaces yields a deeper intuition about why and how our method works. Figure 4d shows a dendrogram where Dravnieks descriptors are arranged according to semantic distance, and color-coded by prediction performance of the *DirSem* model. The prediction's smoothness reveals an odorwheel-like organization

whose backbone is the semantic content of the descriptors: the semantic model for a given descriptor is significantly correlated with the prediction performance of the nearest neighboring descriptor in semantic space ($r = 0.4170$, permutation test $p < 0.001$) and conversely a descriptor's location in semantic space well predicts the prediction performance of the semantic model for that descriptor ($p < 0.001$ measured using 1- and 2-nearest-neighbors permutation tests). On the other hand, the incomplete correspondence between the semantic and olfactory spaces (Fig. 2c, d) is reflected in the failure to incorporate higher-order semantic concepts such as synonymy/antonymy, meronymy/hypernymy, which could be leveraged to improve our model[29].

**Universality and flexibility of the model: Prediction of homologous series**. To demonstrate the universality and flexibility of our zero-shot learning inference, we applied it to odor molecules that have been extensively studied by fragrance chemists and whose structure-odor relationship heuristics are well known. For this, we compiled notes on the smells of 35 molecules containing between two and ten carbon atoms in the homologous series of

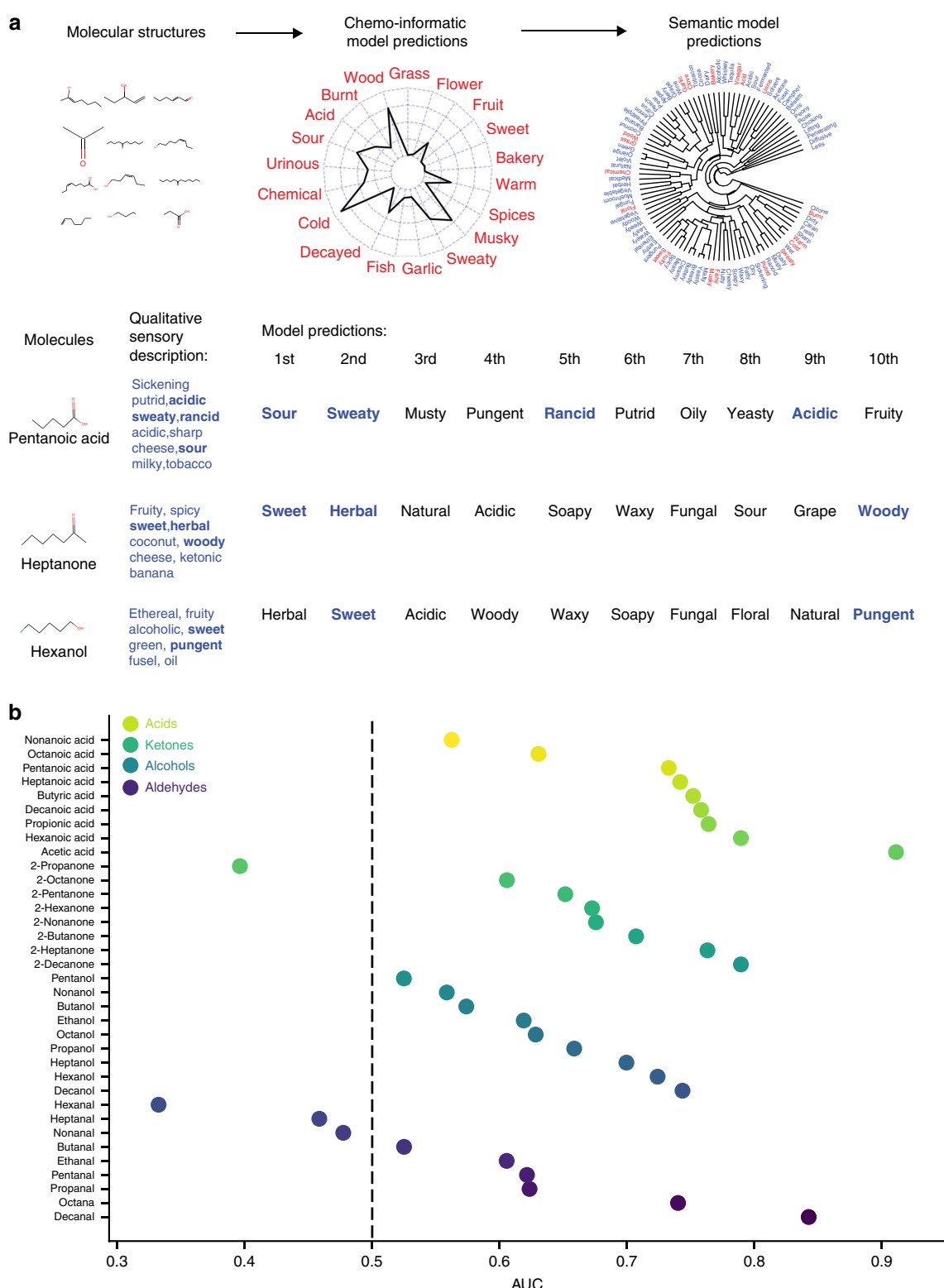

alkyl aldehydes, primary alcohols, 2-ketones, and carboxylic acids (Methods)[30]. For each molecule, using the chemoinformatic and then semantic model method described above, we computed a prediction of the ratings for each of the 80 unique descriptors extracted from the smell notes (Supplementary Data 2). We then ordered for each molecule the descriptors according to their ratings and computed the area-under-the-curve of the

receiver-operating-characteristic curve (AUC) on the binary classification task of predicting whether the paradigm odors for each molecule contains the ordered descriptors (Fig. 5). Notably, the family of acids were the best-predicted family with a median AUC across molecules of 0.75 ($p < 0.02$ one-sided $t$-test), ketones had an AUC of 0.67 ($p < 0.05$ one-sided $t$-test), alcohols had an AUC of 0.63 ($p < 0.07$ one-sided $t$-test) and aldehydes were the

**Fig. 5** Predicting paradigm odors of four molecular families. **a** Schematic for predicting paradigm odors for 35 molecules from the four chemical families of alkyl aldehydes, primary alcohols, 2-ketones, and carboxylic acids. Top. Features from molecular structures, left, are used to predict values of DREAM descriptors middle, and then the semantic model is applied to predict values in 80 unique perceptual descriptors extracted from the paradigm odor descriptions of all the 35 molecules, right. The 80 descriptors are shown in blue in a dendrogram according to their semantic similarity, the 19 DREAM descriptors are shown in red. The 7 overlap descriptors are acid, floral, fruity, sour, sweaty, sweet, and wood. textitBottom. Example of performance of the model for 3 molecules left, their paradigm odors are indicated in blue middle, and predicted ordered list of 80 descriptors by decreasing ratings right, only first 10 are shown. Bold blue descriptors indicate a match to the paradigm descriptors of the molecule. **b** Prediction performance for the paradigm odors for each of the 35 molecules ordered by increasing AUC-ROC values for each of the four families of molecules starting with acids shown with dots in decreasing tones of yellow, ketones in decreasing tones of green, alcohols in increasing tones of light blue and aldehydes in increasing tones of dark blue. The dotted line indicates random AUC-ROC value

worst predicted with an AUC of 0.61 ($p < 0.09$ one-sided $t$-test). The overall median AUC across families of molecules was 0.66 ($p < 0.05$ one-sided $t$-test). Acids were overall predicted as sour but as the number of carbons increased the second-ranked descriptor changed from pungent to sweaty, musty and then back to pungent. Alcohols had overall an herbal smell and changed from sour to sweet (Supplementary Data 2), aldehydes changed from pungent to sweet and fruity, finally 2-ketones changed from sour, acidic to sweet and grape. Here again we encounter the limitations of our current semantic model, reflected in the fact that synonymous odor descriptors have systematically different ranks. For example, each of the 35 molecules is predicted to be more "pungent" than "penetrating" with the median rank of "pungent" being 4 and the median rank of "penetrating" being 54 out of 80.

## Discussion

There is a substantial body of evidence suggesting that the representations of words in semantic vector spaces obtained from co-occurrence statistics can be used to model different aspects of human behavior[31–36]. Distributional semantic models not only provide a good prediction of human word similarity judgments[31,32], but also other psycholinguistic phenomena such as word acquisition in children[33], reaction times in tasks to decide whether a string of letters is a real word or not[37], brain activity as measured via fMRI[34], reading comprehension[35,36], test scores on free-form essays[33], and the presence of cognitive impairments associated with prose recall deficiencies, among others[38].

The present work demonstrates that the general structure of semantic knowledge, as manifested in the unbiased distribution of words in written language, can in fact be mapped onto the olfactory domain, creating a natural classification of olfactory descriptors, an odorwheel, that speaks to the depth of the connection between language and perception[14–19].

This connection can be harnessed to effectively transform ratings from a small set of general descriptors to a larger more specific one. In combination with a chemoinformatics-to-perception model, our work enables end-to-end prediction of perceptual ratings for chemicals for which no ratings data are available at all, that is, a universal predictive map of olfactory perception. Given that specialists including tea and wine tasters, beer brewers, cuisine critics and perfumers expend considerable labor to set up lexicons that are concise and hierarchical, and which cover the relevant odor perception space, a general solution for predicting smell perceptual descriptors, independently of the lexicon used, would be extremely useful across a wide range of industries. Moreover, our findings are also clinically relevant, given that changes in olfactory perception are one of the first signatures of Alzheimer's Disease[39] and associated with a range of other mental disorders[40]. Our approach provides a means to assess directly how these

perceptual disturbances are associated with cognitive and emotional states.

Several limitations of the current approach need to be mentioned, along with possible ways to overcome them. In the first place, the model needs to be extended to mixtures of molecules; a naive linear superposition may suffice, but there is strong evidence that mixtures are particularly susceptible to non-linear interactions[41]. Secondly, as already mentioned, it is possible to enlarge the basic distributional semantic model with additional lexical structure not easily captured by context-as-semantics hypothesis, such as synonyms/antonyms, part-of-speech markers such as verbs and nouns, so as to minimize the distortions we observed in the semantic-to-perception mapping. Related to this last issue, it remains to be seen how the word-based approach presented here will be extended to unconstrained discourse, in particular as it pertains to the expected difference between open narratives of the olfactory-perceptual experience by smell experts and untrained raters[42]. We hope that, for all these extensions, our work will provide a foundation to build upon.

## Methods

**Perceptual data.** In all of our experiments, we predict the average perceptual ratings given to molecules in the Dravnieks human olfaction dataset[23]. This dataset consists of the average ratings of 128 pure molecules by a total of 507 olfaction experts using 146 verbal descriptors. Each molecule was rated only by a subset of 100–150 of the experts. The ratings are on a scale from 0 to 5, where 5 signifies the best match of a descriptor for a given stimulus. Of these 146 descriptors, 15 were discarded because there was no corresponding word vector in our distributional semantic model (e.g., "burnt rubber"), leaving us with 131 descriptors.

Several of our models make use of the data collected by Keller and Vosshall[43] as presented in Keller et al.[24] Data from 49 individuals were used, all of the work reported focuses on predicting the ratings averaged across subjects. Individuals were asked to rate each stimulus using 21 perceptual descriptors (intensity, pleasantness, and 19 descriptors), by moving an unlabeled slider. The default location of the slider was 0. The stimuli were 476 pure molecules. For each task, the final position of the slider was translated into a scale from 0 to 100, where 100 signifies the best match of a descriptor for a given stimulus. Further details on the psychophysical procedures and all raw data are available in the Keller and Vosshall article[43].

**Distributional semantic model.** To assess accurately the semantic similarity between the DREAM and Dravnieks descriptors, we took advantage of a distributional semantic model trained using the fastText skip-gram algorithm, a neural network-based model that predicts word occurrence based on context[44]. These 300-dimensional vectors were trained on a corpus of 16 billion words, and are publicly available (https://fasttext.cc/docs/en/english-vectors.html). See Bojanowski et al. for additional details on training and the specifics of the model. We originally used Word2Vec vectors[26], and though we saw improved performance with fastText, the difference was quite small.

The semantic vectors of a distributional semantic model are vectorial representations of word meaning motivated by the distributional hypothesis stating that the meaning of a word can be inferred as a function of the linguistic contexts in which it occurs[25].

Distributional semantic models rest on the assumption that, to quote Wittgenstein, that 'the meaning of a word is its use in the language'[45]. For example, the distributional hypothesis would predict that *kitten* and *cat* have similar meanings, given that they are both used in contexts such as the _____ *purred softly* and the _____ *licked its paws*; meanwhile the meaning of *rock* would be less similar

to *kitten*, because it is rarely if ever used in similar contexts. The distributional hypothesis has inspired the field of *distributional semantics*, which aims to quantify the meanings of words based on co-occurrence statistics of the words in large samples of written or spoken language. These co-occurrence statistics can be summarized and embedded in a low-dimensional vector space, known as a *semantic vector space*, using dimensionality-reduction techniques such as principal components analysis[46] or neural networks[26]. The semantic vector space is constructed in such a way that words that occur in similar contexts and are therefore presumably semantically similar are represented by vectors that are geometrically close as measured for example by cosine distance or Euclidean distance.

**Chemoinformatic features**. We used version six of the Dragon software package (http://www.talete.mi.it) to generate a 4884 physicochemical features of each molecule (including atom types, functional groups, topological, and geometric properties)

**Estimating the perceptual ratings from chemical structure**. To estimate the perceptual ratings from the chemical structure, we use a regularized linear model that is learned using elastic net regression[24]. This model is trained on the DREAM dataset of 476 molecules. The input for the model consists of the chemoinformatic features of the molecules described above. Using these features, the model predicts the mean perceptual rating given by 49 subjects on each of the perceptual descriptors that we use above. Thus, for each molecule $i$, the chemoinformatics-to-perception model learns a transformation $\mathbf{C}$ such that

$$\widehat{\mathbf{p}}_{S,i} = \mathbf{C}\mathbf{x}_{S,i}, \tag{1}$$

where $\widehat{\mathbf{p}}_{S,i}$ is the 19-dimensional vector containing the model's estimate of the mean ratings on the DREAM descriptors for the molecule $i$, and $\mathbf{x}_{S,i}$ is the 4884-dimensional vector of molecule $i$'s chemoinformatic features.

**Extending ratings to new descriptor lexicons**. We define two tasks, direct and imputed. For the direct task, we have access to actual DREAM ratings for each test molecule. In the imputed task, we do not have access to the test molecule's actual DREAM ratings. Instead, we begin by applying a previously trained and unpublished model used in the context of Keller et al.[2] that can infer the ratings scores of any chemical on the DREAM verbal descriptors, given its chemoinformatic properties. For both tasks, the objective is to predict the test molecule's Dravnieks ratings. Consequently, we also refer to the DREAM data as our *source* and the Dravnieks data as our *target*. We present three classes of model for each task, ratings, semantic, and mixed. Altogether the combination of the tasks and model classes results in six models, which we describe below.

As before, the real or imputed DREAM ratings scores for each molecule $i$ can be collected into a 19-dimensional perceptual vector $\mathbf{p}_{S,i}$. In addition, for each DREAM descriptor $d$, we have a semantic vector $\mathbf{s}_{S,d}$, which is a 300-dimensional vector computed as described in the section describing the semantic vectors. We collect these into a *source semantic matrix* $\mathbf{S}_S$ of dimension $19 \times 300$ where again 19 is the number of DREAM perceptual descriptors.

We want to learn the ratings scores for any arbitrary set of descriptors—we call these our *target* descriptors. We assume that we can compute the semantic vectors corresponding to each of these perceptual descriptors $d$, denoted by $\mathbf{s}_{T,d}$. Taking advantage of the structure inherent in these target semantic vectors is key to our method. We collect these into a *target semantic matrix* $\mathbf{S}_T$ of dimension $D_T \times 300$ where $D_T$ is the number of target descriptors. In the case of the results presented in the body of this paper, $D_T = 131$, because there are 131 Dravnieks descriptors that we use.

In this framework, our goal is to estimate the ratings scores for the target (Dravnieks) descriptors for each test molecule $i$, denoted by $\mathbf{p}_{T,i}$.

In order to set a point for comparison, we propose a baseline model that takes the mean rating score for each target-set descriptor, across the training set of molecules for which ratings are available:

$$\overline{\mathbf{p}}_T = \frac{1}{|\text{Training Set}|} \sum_{i \in \text{Training Set}} \mathbf{p}_{T,i} \tag{2}$$

This is then used as the baseline estimate of the ratings scores across the target-set descriptors for a given new test molecule $i$. In the case where no training ratings are available for the target descriptors, we take the baseline to be the constant vector $\mathbf{0}$.

The first model class is composed of the semantics-only models for the direct and imputed tasks (*DirSem* and *ImpSem*, respectively). These semantics-only models assume that a distributional semantic space derived from a linguistic corpus shares structure with the olfactory-perceptual space in which perceptual ratings scores exist. Consequently, we seek to test whether we can leverage the structure of the semantic space to predict ratings in the perceptual ratings space. To learn the semantics-only model $\mathbf{S}$ we proceed by supposing there exists a matrix $\mathbf{S}$ of dimension $19 \times 131$ that roughly maps from the semantic vectors for the source set of perceptual descriptors to the semantic vectors (collected into the matrix $\Sigma_S$) for

the target set of perceptual descriptors (collected into the matrix $\Sigma_T$:

$$\Sigma_T \approx \mathbf{S}'\Sigma_S. \tag{3}$$

Our semantics-only models make the assumption that $\mathbf{S}$ is also an appropriate transformation for mapping from the perceptual ratings for the source set of descriptors to the perceptual ratings for the target set for each molecule $i$:

$$\mathbf{p}_{T,i} \approx \mathbf{S}\mathbf{p}_{S,i}. \tag{4}$$

In order to estimate $\mathbf{S}$, we use elastic net regression. The regularization parameters are set by nested 10-fold cross-validation.

Note that of the three model types described in this section, the semantics-only models are the only ones that do not rely on having access to any ratings scores for the source set (i.e., no $\mathbf{p}_T$ is required for training). However, to compare this model directly with the models that do use such information, we tested the effect of adding information about the mean rating to the model. Therefore, the final estimate for molecule $i$ under this model, when target descriptors training molecules are available, would be:

$$\widehat{\mathbf{p}}_{T,i} = \mathbf{S}\mathbf{p}_{S,i} + \overline{\mathbf{p}}_T. \tag{5}$$

The only difference between *DirSem* and *ImpSem* is in the nature of $\mathbf{p}_{S,i}$ and $\overline{\mathbf{p}}_T$. Recall that in *DirSem* these are derived from real DREAM ratings data, while in *ImpSem* they are predictions of the chemoinformatics-to-perception model.

The ratings-only models *DirRat* and *ImpRat* rely on having access to ratings scores for the target descriptors, for some training set of molecules. They assumes that there is some function $\mathbf{R}$ that maps from ratings scores on the source descriptors to ratings scores on the target descriptors for each molecule $i$:

$$\mathbf{p}_{T,i} \approx \mathbf{R}\mathbf{p}_{S,i}. \tag{6}$$

Once again, we estimate $\mathbf{R}$ using elastic net regression, with regularization weights set by nested 10-fold cross-validation. We also add information about the mean rating to the model, if available, so our final estimate under this model is:

$$\widehat{\mathbf{p}}_{T,i} = \mathbf{R}\mathbf{p}_{S,i} + \overline{\mathbf{p}}_T. \tag{7}$$

For the mixed models direct and imputed we simply average the predictions of the semantics-only and ratings-only models:

$$\widehat{\mathbf{p}}_{T,i} = \frac{1}{2}(\mathbf{R} + \mathbf{S})\mathbf{p}_{S,i} + \overline{\mathbf{p}}_{\mathbf{T}}. \tag{8}$$

In preliminary investigations, we also looked at other ways to combine the information in the semantics-only and ratings-only models, such as training a single regression model on the set union of the descriptors' semantic vector values and molecule ratings, but a simple average performed best.

**Evaluating performance**. For each model, we vary the number of training molecules for which target descriptor ratings are available. We can then measure the median Pearson correlation between model $M$'s estimate $\widehat{\mathbf{p}}_{T,i}$ and the ground truth $\mathbf{p}_{T,i}$ for each test molecule $i$ as:

$$r_{M,G}^{(i)} = \left\{ r\left(\widehat{\mathbf{p}}_T^{(i)}, \mathbf{p}_{T,i}\right) \right\}. \tag{9}$$

We use these correlations to assess whether the model's performance differs significantly from the baseline model, by computing $Z$-scores. For the Semantics-Only model when we do not use any training molecules, the baseline is simply a correlation of zero, so the $Z$-score can be obtained using the Fisher $r$-to-$Z$ transformation:

$$Z_{M,G,i} = \frac{1}{2}\log\left(\frac{1 + r_{M,G,i}}{1 - r_{M,G,i}}\right). \tag{10}$$

However, for the other models, note that the correlation coefficient produced by model and the correlation coefficient produced by the Baseline model are not independent random variables. Thus, to determine whether these two correlations differ significantly, we must take their dependence into account, which the standard Fisher transformation does not do. Instead, we can use the method developed by ref.[47]:

$$Z_{M,G,i} = \sqrt{N-3}\frac{Z_{G,B,i} - Z_{M,B,i}}{\sqrt{2(1-s)}}, \tag{11}$$

where

$$s = \frac{r_{G,B,i}\left(1 - r_{G,M}^2 - r_{M,B,i}^2\right) - \frac{1}{2}(r_{G,M,i}r_{M,B,i})\left(1 - r_{G,M,i}^2 - r_{M,B,i}^2 - r_{G,B,i}^2\right)}{\left(1 - r_{G,M,i}^2\right)\left(1 - r_{M,B,i}^2\right)}. \quad (12)$$

We can then compute the median of these $Z$-scores for all molecules in the test set:

$$\underset{i \in \text{Test Set}}{\text{median}} \; Z_{M,G,i}$$

**Permutation tests for evaluating smoothness in semantic prediction**. We performed a permutation test by randomly permuting the semantic nearest neighbors of each descriptor, and then re-computing the correlation between the prediction performances (measured by Pearson's correlation) of each point and of its permuted nearest neighbor. The resulting simulated correlations exceeded the true correlation of $r = 0.4170$ in 0 of the 10,000 permutations.

For each descriptor, the $k$-nearest neighbor ($k$-NN) algorithm predicts the descriptor's prediction performance (measured by Pearson's $r^2$) by taking the distance-weighted average of the prediction performance of the $k$-nearest neighbors. The mean squared error of this algorithm is then computed, and the significance is evaluated using a permutation test. The permutation test is performed by randomly permuting the semantic nearest neighbors of each descriptor, and then re-computing the mean squared error of the resulting $k$-NN predictions. The mean squared error of 2000 such permutations was never below that of the true mean squared error.

**Tests for similarity between ratings and semantic vectors correlation matrices**. To estimate the degree of structural similarity between the correlation matrix defined by Dravnieks and DREAM ratings (Fig. 2a), and that defined by the corresponding semantic vectors (Fig. 2b), we implemented two tests. In the first one, we computed the Procrustes dissimilarity between the rating matrix and the semantic matrix, and compared it against the expected dissimilarity between the original rating matrix and random permutation surrogates of the semantic matrix. A Wilcoxon test yields $p < 0.05$. For the second test, we found for each DREAM descriptor the Dravnieks descriptor with which it is maximally correlated, both in the ratings and semantic matrices. A Spearman test for the correlation between these two sequences yields $r = 0.74$, $p < 10^{-4}$. Conversely, the test for the maxima estimated along the Dravnieks descriptors yields $r = 0.5$, $p < 10^{-9}$.

**Organization of semantic and rating spaces**. The dendrogram in Fig. 4d was created by computing the cosine distance between the semantic vectors of the Dravnieks descriptors, fed into an agglomerative hierarchical cluster tree algorithm using the average over all element of a cluster to determine the distance between clusters (*linkage* function[48]). The 2D projections in Fig. 2c and d were created using multidimensional scaling (*mdscale* function[48]) with cosine distance for both maps. For the semantic organization, the fastText 300-dimensional vectors corresponding to the DREAM descriptors were used; for the ratings organization, each descriptor was represented as vector of ratings over molecules.

**Additional information on elastic net regression**. LASSO and elastic net are regression algorithms that impose a regularization penalty on the regression weights in order to reduce model complexity and avoid overfitting.

For a regression model of the form

$$\mathbf{Y} = \mathbf{AX}, \quad (13)$$

the regression weights in LASSO are estimated in order to minimize the following loss function:

$$\sum_i \left\|\mathbf{Y}_i - (\mathbf{AX})_i\right\|_2^2 + \lambda_1 \sum_i \|\mathbf{A}_i\|_1, \quad (14)$$

where the first term is the squared error of the prediction, and the second term is a regularization penalty (a penalty on the regression weights), and $\lambda_1$ is a regularization strength parameter. LASSO's regularization penalty leads to a model that is sparse (i.e., produces few nonzero regression weights). This results in relatively more parsimonious and interpretable model. However, the LASSO loss function is not convex, so it does not produce a unique solution when the number of features is greater than the number of samples. When two features are highly correlated, LASSO will arbitrarily assign only one of the two features a nonzero weight, even if both contribute equally to the prediction in the ground truth model. This can lead to poor prediction performance.

Elastic net regression attempts to get around LASSO's drawbacks. The regression weights are computed according to

$$\hat{\mathbf{A}} = \underset{\mathbf{A},\lambda_1,\lambda_2}{\arg\min} \sum_i \|\mathbf{Y}_i - \mathbf{AX}_i\|_2^2 + \lambda_1 \sum_i \|A_i\|_1 + \lambda_2 \sum_i \|A_i\|_2^2, \quad (15)$$

where the first term is the squared error of the prediction, the second term is the L1 (or LASSO) regularization penalty, the third term is the L2 (or ridge regression) regularization penalty[49], and $\lambda_1$ and $\lambda_2$ are the corresponding regularization strengths. Elastic net regression seeks to combine the benefits of LASSO and ridge regression. Like LASSO, it results in a parsimonious, interpretable, sparse model where most of the regression coefficients are zero. However, like ridge regression, elastic net has a convex loss function and produces a unique solution even when the number of features is greater than the number of samples. Elastic net also overcomes the arbitrary feature selection drawback of LASSO. See[49] for more details.

**Sequentially selecting prototypical features**. We now describe the technical details of the method used to create Fig. 3a. For a more thorough treatment please refer to ref.[28]

Let $\mathcal{X}$ be the space of all covariates from which we obtain the samples $X^{(1)}$ and $X^{(2)}$; in our particular case, these will be the perceptual ratings over molecules. Consider a kernel function $k : \mathcal{X} \times \mathcal{X} \to \mathbb{R}$ and its associated reproducing kernel Hilbert space (RKHS) $\mathcal{K}$ endowed with the inner product $k(\mathbf{x}_i, \mathbf{x}_j) = \langle \phi(\mathbf{x}_i), \phi(\mathbf{x}_j) \rangle$ where $\phi_{\mathbf{x}}(\mathbf{y}) = k(\mathbf{x}, \mathbf{y}) \in \mathcal{K}$ is continuous linear functional satisfying $\phi_{\mathbf{x}}:h \to h(\mathbf{x}) = \langle \phi_{\mathbf{x}}, h \rangle$ for any function $h \in \mathcal{K} : \mathcal{X} \to \mathbb{R}$.

The maximum mean discrepancy (MMD) is a measure of difference between two distributions $p$ and $q$ where if $\boldsymbol{\mu}_p = \mathbb{E}_{\mathbf{x} \sim p}[\phi_{\mathbf{x}}]$ it is given by:

$$\text{MMD}(\mathcal{K}, p, q) = \sup_{h \in \mathcal{K}} \left(\mathbb{E}_{\mathbf{x} \sim p}[h(\mathbf{x})] - \mathbb{E}_{\mathbf{y} \sim q}[h(\mathbf{y})]\right)$$

$$= \sup_{h \in \mathcal{K}} \langle h, \mu_p - \mu_q \rangle.$$

Our goal is to approximate $\boldsymbol{\mu}_p$ by a weighted combination of $m$ sub-samples $Z \subseteq X^{(2)}$ drawn from the distribution $q$, i.e., $\boldsymbol{\mu}_p(\mathbf{x}) \approx \sum_{j:\mathbf{z}_j \in Z} w_j k(\mathbf{z}_j, \mathbf{x})$ where $w_j$ is the associated weight of the sample $\mathbf{z}_j \in X^{(2)}$. We thus need to choose the prototype set $Z \subseteq X^{(1)}$ of cardinality ($|.|$) $m$ where $n^{(1)} = |X^{(1)}|$ and learn the weights $w_j$ that minimizes the finite sample *MMD* metric with the additional *non-negativity constraint* for interpretability, as given below:

$$\widehat{\text{MMD}}(\mathcal{K}, X^{(1)}, Z, \mathbf{w})$$
$$= \frac{1}{(n^{(1)})^2} \sum_{\mathbf{x}_i, \mathbf{x}_j \in X^{(1)}} k(\mathbf{x}_i, \mathbf{x}_j) - \frac{2}{n^{(1)}} \sum_{\mathbf{z}_j \in Z} w_j \sum_{\mathbf{x}_i \in X^{(1)}} k(\mathbf{x}_i, \mathbf{z}_j)$$
$$+ \sum_{\mathbf{z}_i, \mathbf{z}_j \in Z} w_i w_j k(\mathbf{z}_i, \mathbf{z}_j); \text{subject to } w_j \geq 0, \forall \mathbf{z}_j \in Z. \quad (16)$$

Index the elements in $X^{(2)}$ from 1 to $n^{(2)} = |X^{(2)}|$ and for any $Z \subseteq X^{(2)}$ let $L_Z \subseteq [n^{(2)}] = \{1, 2, \ldots, n^{(2)}\}$ be the set containing its indices. Discarding the constant terms in (16) that do not depend on $Z$ and $\mathbf{w}$ we define the function

$$l(\mathbf{w}) = \mathbf{w}^T \boldsymbol{\mu}_p - \frac{1}{2} \mathbf{w}^T K \mathbf{w} \quad (17)$$

where $K_{ij} = k(\mathbf{y}_i, \mathbf{y}_j)$ and $\mu_{p,j} = \frac{1}{n^{(1)}} \sum_{\mathbf{x}_i \in X^{(1)}} k(\mathbf{x}_i, \mathbf{y}_j); \forall \mathbf{y}_j \in X^{(2)}$ is the point-wise empirical evaluation of the mean $\boldsymbol{\mu}_p$. Our goal then is to find an index set $L_Z$ with $|L_Z| \leq m$ and a corresponding $\mathbf{w}$ such that the set function $f : 2^{[n^{(2)}]} \to \mathbb{R}^+$ defined as

$$f(L_Z) \equiv \underset{\mathbf{w} : \text{supp}(\mathbf{w}) \in L_Z, \mathbf{w} \geq 0}{\max} l(\mathbf{w}) \quad (18)$$

is maximized. Here $\text{supp}(\mathbf{w}) = \{j : w_j > 0\}$. We will denote $\boldsymbol{\zeta}^{(L_Z)}$ the maximizer of set $L_Z$.

The above problem is NP-hard to solve. The ProtoDash algorithm, however, efficiently solves this problem and is shown to have a tight approximation guarantee[28]. If $Q$ denotes the $476 \times 19^{24}$ perceptual matrix then we set $X^{(1)} = X^{(2)}$

$= Q^T$ and run the following algorithm.

---

**Algorithm 1** ProtoDash

**Input:** $X^{(1)}, X^{(2)}$
$L = \emptyset, \boldsymbol{\zeta}^{(L)} = \mathbf{0}$
$\mathbf{g} = \nabla l(\mathbf{0}) = \boldsymbol{\mu}_p$
$i = 1$
**while** $i \leq 19$ **do**
 $j_0 = \underset{j \in \lceil n^{(2)} \rceil \setminus L}{\operatorname{argmax}} g_j$
 $L = L \cup \{j_0\}$
 $\boldsymbol{\zeta}^{(L)} = \underset{\mathbf{w}:supp(\mathbf{w}) \in L, \mathbf{w} \geq 0}{\operatorname{argmax}} l(\mathbf{w})$ $\{l(.)$ depends on $X^{(1)}$ and $X^{(2)}.\}$
 $\mathbf{g} = \nabla l\left(\boldsymbol{\zeta}^{(L)}\right) = \boldsymbol{\mu}_p - K\boldsymbol{\zeta}^{(L)}$
 $i = i + 1$
**end while**
**return** $L, \boldsymbol{\zeta}^{(L)}$

---

The order in which elements are added to $L$ is the order depicted in Fig. 3a.

**Predictions of paradigm odors for molecular families**. We extracted every term used to describe the paradigm odors for any of the 35 molecules in 4 families: 9 molecules from the family of alkyl aldehydes, 9 molecules from primary alcohols, 8 molecules from 2-ketones, and 9 molecules from carboxylic acids, that appeared in *The Good Scents Company* and *Perfumer and Flavorist* libraries. We included 80 descriptors used to describe all the 35 molecules, ignoring instances where the term was only weakly associated–e.g., *fruity nuance* or *weak hint of apple*. We then predicted for these 35 molecules the 19 DREAM perceptual descriptors from the Dragon molecular descriptors of the molecules and then used the Semantic model to obtain ratings for the 80 terms. Besides the AUC, we also computed for each molecule a *p*-value by performing a one-sided *t*-test for the difference between the means of the predictions for the terms that were used to describe the molecule and the terms that were not used to describe the molecule. A Kolmogorov–Smirnov test on these *p*-values reveals that they are not uniformly distributed ($p < 1e–6$), hence, overall predicted ratings for descriptors that are used to for a molecule rank much higher than the ones that are not.

## Data availability

All relevant data are available from the authors. The code to predict DREAM descriptors is available here: https://github.ibm.com/adhuran/Olfaction The code to predict Dravnieks descriptors is available here: https://github.com/edg2103/odormatic.

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

## Acknowledgements

We would like to thank Rick Gerkin and Pablo Polosecki for reading the manuscript and providing useful comments. Smell Icon used was adapted from Taka Oumehara from https://thenounproject.com and book icon from Zlatko Najdenovski from www.flaticon.com.

## Author contributions

E.D.G. developed the predictions and semantic model, A.D. developed the chemoinformatic model. All authors together interpreted the results, approved the design of the figures and the text, which were prepared by E.D.G., G.A.C., and P.M.

## Additional information

**Competing interests:** The authors declare no competing interests.

