## [Peer Review File · Nature Communications]

Reviewers' comments:

Reviewer #1 (Remarks to the Author):

Predicting natural language of smells.

The paper presents several interesting contributions related to semantic analysis of mono-molecular odorants. In particular, the authors show that words embeddings can be effectively used to define models able to predict perceptual ratings over a potentially large and arbitrary set of descriptors. Such models, enriched with additional features (e.g. compound features), can further be used to predict perceptual rating of arbitrary molecules. These models can also be used to distinguish domain-specific sets of descriptors and odor wheels.

The contributions are validated using various datasets and statistical tests.

The current version of the paper is difficult to understand; it could be substantially improved by:

- Clarifying the aim of the study. The authors are studying various complex problems in the same paper, and those problems are only implicitly mentioned: (i) how to distinguish domain-specific sets of descriptors relevant to odor analysis, and how to structure them to automatically build odor wheels, (ii) how to predict perceptual ratings based on prior knowledge about existing ratings and descriptor relationships (through ratings), (iii) how to address the main goal aiming to automatically rate unknown odorants.

- Detailing what perceptual descriptor rating means. The authors use that notion early in the paper without defining it, e.g. "To investigate whether semantic representations derived from language use could be applied to reliably predict ratings of a large set of detailed olfactory perceptual descriptors [...]". It is however difficult for the reader to understand that they are referring to ratings of descriptors considering specific molecules.

- Detailing and modifying the Method section. Details should be added in order to clarify specific points (detailed hereafter).

- Extending future work sections. It could be interesting to provide an overview of the work to be done in order to be able to fully predict natural language description of smells. Authors' contributions should also be discussed according to it.

- Provide a related work section: e.g. cite other contributions related to semantic analysis of odor (also using word embeddings), e.g. Medjkoune et al. 2016 Towards a Non-oriented Approach for the Evaluation of Odor Quality. Proceedings, Part I of the 16th International Conference, IPMU 2016, Eindhoven, The Netherlands, June 20-24, 2016, ISSN: 1865-0929. DOI:10.1007/978-3-319-40596-4_21

Other recommendations and comments:

- Modify the title of the paper; it is not in accordance with the contribution – smells are most often more complex than mono-molecular odorants.

- The benefits and the impact of the contribution introduced in the conclusion (p8) should be presented in the introduction to motivate the study.

- The correspondence between perceptual and semantic spaces should be further studied. Differences between those spaces could have been illustrated to underline the difficulty of the task, e.g. by showing distortions (regions that differ strongly in term of distances). Additionally, considering provided title, advanced discussions related to the used of more complex linguistic units such as simple nominal groups are expected.

- Discussion related to the limits of the approach should be extended.

- The source code used to perform the experiments should be publicly available in order to be able to reproduce the results.
- Performance of models should be discussed considering inter-agreement of experts.
- It is not clear why the authors are using a correlation instead of a more classical quadratic error to evaluate the approaches.
- It could be interesting to comment the impact of the choice of the method – and (hyper)-parameters- used to build embeddings (word2vec, Glove, using various parameters such as different vector sizes). Word embeddings are the cornerstone of the contribution; such technical results are important in this context.
- More complex non-linear state-of-the-art models could have been used (non-linear SVM) and should be tested. The fact that only linear models have been tested should at least be motivated.
- Please detail what is the similarity measure used to compare the embeddings (cosine similarity and Euclidian are mentioned, the one used in practice is not specified).
- The protoDash algorithm should be detailed or removed.

Minor:

- p8 out of 80
- p8 can be used to model human behaviour [of ...]. Complete.
- p23 neural networks are not commonly mentioned as dimensionality reduction techniques such as ACP, (even if they implicitly rely on it, or can be used to do so, e.g. auto-encoder).
- p25 Word2vec – the method used in the final version is fastText.
- p26 takes take
- p28 3rd equation parenthesis missing
- p33 citeproto missing citation

Reviewer #2 (Remarks to the Author):

Title: Predicting natural language descriptions of smells

Summary: The manuscript described above shows the method to predict odor impression using natural language semantic representation. When chemoinformatic features such as molecular features for each molecule are given, the ratings of odor descriptors can be predicted. The authors mentioned the semantic-based approach achieved high accuracy to predict perceptual ratings of smell.

Although the level of the manuscript is relatively high, the acceptance of this manuscript as a paper of Nature Communications is not recommended because of its originality. The following paper using natural language semantic representation to predict impression of odor has been already published although the input data for prediction are different.

Predictive modeling for odor character of a chemical using machine learning combined with natural language processing, Yuji Nozaki and Takamichi Nakamoto, PLoS ONE 13(6): e0198475. <https://doi.org/10.1371/journal.pone.0198475>, June 14, 2018.

Since the main point of this work is to use natural language semantic representation to predict odor impression, its novelty is weak for Nature communication. It should represent an advance which is likely to influence thinking in the field.

However, I think that the manuscript includes several new things and ideas acceptable for other journal if the authors modify their claims. I suggest that the authors can reorganize the

manuscript and submit it to other journal. The following comments might be helpful to reorganize the manuscript.

Figure 1

The reason for using two data sets such as Dream and Dravnieks ones are not clearly described. It seems better to use one unified data set. Since the authors used two data sets, the data flow became complicated. Moreover, it is not clear whether the accuracy to convert from Dream ratings to Dravnieks ratings is sufficient.

How many descriptors are necessary to express an odor impression? If 19 descriptors are enough, why 131-descriptor data set was used together? If 131 descriptor-approach is the best way, 19-descriptor data set lacks some information.

Figure 2 a

It is not clear how direct semantic method is used to predict Dravnieks rating. Transformation S maps semantic descriptor vector of dream onto the semantic descriptor vector of Dravnieks. It does not mean the transformation of dream rating vector into Dravnieks rating vector.

Figure 2 a bottom

Although the authors claim that the direct semantic model uses no molecule during training, why does the lateral axis mean number of Dream/Dravnieks overlap training molecules.

Figure 2 a,b bottom

In my understanding, the authors claim the advantage of direct semantic model. However, direct rating model is always better than direct semantic model when the number of training molecules was large. Thus, a user should use direct rating model together with large number of training molecules.

Figure 3 a

Although the correlation increases as the number of descriptors increases, the value of correlation seems low. The authors should comment on this point.

Figure 3 b, c

The authors might claim that mixed model is better than direct semantic model. However, the meaning of mixed model is the same as that of direct rating model since it uses molecule information. The authors should comment on this point.

In abstract (Page 2), the authors mentioned that establishing that the semantic distance between descriptors defines the equivalent of an odorwheel.

In Figure 3 d, there are many descriptors with low correlation across molecules. Can the authors say that they have established the method? Otherwise did they just find the possibility? The author should describe the level of completion.

Figure 4

Why not all but 80 descriptors were used here? The authors mentioned 7 overlap descriptors. What groups of descriptors share 7 descriptors?

Figure 4 b

Where are qualitative sensory description coming from? Is it possible to show its rating in comparison with true values?

Reviewer #3 (Remarks to the Author):

This is a rigorous and interesting application of machine learning to an important question about odor perception. Congrats to the authors on a novel and nicely executed study.

Describing smells with language is notoriously difficult and open ended, and begs the question of how much our verbal labels for odors are anchored by olfactory experience vs. 'simply' reflective of the much broader semantic life and relationships that words have among themselves. In other words, if someone had a crummy sense of smell, but a great vocabulary, could they still describe odors using all the "right" words? This study says yes, and that is an important result.

I don't have any major technical concerns and found the work to be thorough. I think it would be an important addition to the literature that would arouse broad interest.

Major comments:

1) I'm not sure I arrive at the same conclusion from the comparison between Fig. 1b and 1c that the authors do. Namely, that the two correlation matrices are similarly structured, and given that, "favor the hypothesis of a tight perceptual-linguistic bond between the descriptors ratings and their linguistic meanings." If that were the case, shouldn't we see some of the same block structure in 1c as in 1b? Instead, 1c seems rather sparse, with only a few strong, singular hits showing up in those places where the semantic descriptor vectors will be identical. I know the significance tests say the matrices are similar, but I wonder if that could be driven by a relatively small subset descriptors for which the "tight link" hypothesis holds very strongly, while the hypothesis is much less compelling across the whole of descriptor space. (More below).

2) Related: from 1c, the only case for which I can convince myself that there's some broad correspondence between DREAM semantic descriptors and Dravnieks descriptors is for the {Flower, Sweet, Fruit} block that the the authors emphasize. One could imagine that there are a subset of words that, when used, tend to be used in contexts where one is describing odors. Many other words, however, might have their meaning distributed across many additional contexts, and are only occasionally co-opted to describe odors. Did you/could you look to see whether dropping certain specific descriptors (or combinations) has much more dramatic consequences for your models than others? It would be useful to know if dropping {'Flower' and 'Sweet'}, for example, has consequences that were little distinguishable from dropping any other pair descriptors.

Minor comments:

The 'DREAM ratings' and 'Dravnieks ratings' schematics you show in figure 2 are confusing, and I'm not sure they do much to clarify what you're doing. I had to zoom in 250% before I could really see that they were supposed to represent ratings sliders. I'd recommend trying to come up with a more compelling visual.

P14, legend: "...and they have 58 molecules are common to both data sets." → "... they have 58 molecules in common to both..."

Reviewer 1:

Recommendation:

Comments:

Predicting natural language of smells.

The paper presents several interesting contributions related to semantic analysis of mono-molecular odorants. In particular, the authors show that words embeddings can be effectively used to define models able to predict perceptual ratings over a potentially large and arbitrary set of descriptors. Such models, enriched with additional features (e.g. compound features), can further be used to predict perceptual rating of arbitrary molecules. These models can also be used to distinguish domain-specific sets of descriptors and odor wheels.

The contributions are validated using various datasets and statistical tests.

The current version of the paper is difficult to understand; it could be substantially improved by:

- Clarifying the aim of the study. The authors are studying various complex problems in the same paper, and those problems are only implicitly mentioned: (i) how to distinguish domain-specific sets of descriptors relevant to odor analysis, and how to structure them to automatically build odor wheels, (ii) how to predict perceptual ratings based on prior knowledge about existing ratings and descriptor relationships (through ratings), (iii) how to address the main goal aiming to automatically rate unknown odorants.

- Detailing what perceptual descriptor rating means. The authors use that notion early in the paper without defining it, e.g. “To investigate whether semantic representations derived from language use could be applied to reliably predict ratings of a large set of detailed olfactory perceptual descriptors [...]”. It is however difficult for the reader to understand that they are referring to ratings of descriptors considering specific molecules.

- Detailing and modifying the Method section. Details should be added in order to clarify specific points (detailed hereafter).

- Extending future work sections. It could be interesting to provide an overview of the work to be done in order to be able to fully predict natural language description of smells. Authors' contributions should also be discussed according to it.

- Provide a related work section: e.g. cite other contributions related to semantic analysis of odor (also using word embeddings), e.g. Medjkoune et al. 2016 Towards a Non-oriented Approach for the Evaluation of Odor Quality. Proceedings, Part I of the 16th International Conference, IPMU 2016, Eindhoven, The Netherlands, June 20-24, 2016, ISSN: 1865-0929. DOI:10.1007/978-3-319-40596-4_21

Other recommendations and comments:

- Modify the title of the paper; it is not in accordance with the contribution – smells are most often more complex than mono-molecular odorants.
- The benefits and the impact of the contribution introduced in the conclusion (p8) should be presented in the introduction to motivate the study.
- The correspondence between perceptual and semantic spaces should be further studied. Differences between those spaces could have been illustrated to underline the difficulty of the task, e.g. by showing distortions (regions that differ strongly in term of distances). Additionally, considering provided title, advanced discussions related to the used of more complex linguistic units such as simple nominal groups are expected.
- Discussion related to the limits of the approach should be extended.
- The source code used to perform the experiments should be publicly available in order to be able to reproduce the results.
- Performance of models should be discussed considering inter-agreement of experts.
- It is not clear why the authors are using a correlation instead of a more classical quadratic error to evaluate the approaches.
- It could be interesting to comment the impact of the choice of the method – and (hyper)-parameters- used to build embeddings (word2vec, Glove, using various parameters such as different vector sizes). Word embeddings are the cornerstone of the contribution; such technical results are important in this context.
- More complex non-linear state-of-the-art models could have been used (non-linear SVM) and should be tested. The fact that only linear models have been tested should at least be motivated.
- Please detail what is the similarity measure used to compare the embeddings (cosine similarity and Euclidian are mentioned, the one used in practice is not specified).
- The protoDash algorithm should be detailed or removed.

Minor:

- p8 out of 80
- p8 can be used to model human behaviour [of ...]. Complete.
- p23 neural networks are not commonly mentioned as dimensionality reduction techniques such as ACP, (even if they implicitly rely on it, or can be used to do so, e.g. auto-encoder).
- p25 Word2vec – the method used in the final version is fastText.
- p26 takes take
- p28 3rd equation parenthesis missing
- p33 citeproto missing citation

Response to Reviewer 1 (general): [We appreciate the support the reviewer expressed for the results presented in the manuscript, and the generous feedback provided. We modified in consequence the manuscript and believe it has improved significantly its appeal to the general readership of Nat. Comms. A detailed answer to the comments follows.]

Response to each comment of Reviewer 1 (individually):

- Clarifying the aim of the study. The authors are studying various complex problems in the same paper, and those problems are only implicitly mentioned: (i) how to distinguish domain-specific sets of descriptors relevant to odor analysis, and how to structure them to automatically build odor wheels, (ii) how to predict perceptual ratings based on prior knowledge about existing ratings and descriptor relationships (through ratings), (iii) how to address the main goal aiming to automatically rate unknown odorants.

[We thank the reviewer for these comments that boil down the essence of the paper. We made our abstract more concise and expanded our introduction in order to clarify the points raised by the reviewer. We also divided our paper into sections and subsections to further clarify the aim of the study and of the individual experiments, and added a new figure.]

- Detailing what perceptual descriptor rating means. The authors use that notion early in the paper without defining it, e.g. “To investigate whether semantic representations derived from language use could be applied to reliably predict ratings of a large set of detailed olfactory perceptual descriptors [...]”. It is however difficult for the reader to understand that they are referring to ratings of descriptors considering specific molecules.

[We agree with the reviewer that this was not clear enough in our earlier version, and we have gone into more detail on this concept in the second paragraph of the Introduction while also adding an explanatory figure (Fig.1). We also changed the text from the Results section that the reviewer quoted to read as follows:]

To investigate whether semantic representations derived from language use could be applied to reliably predict how molecules are rated along a large set of detailed olfactory perceptual descriptors, we chose to predict the ratings of 146 fine-grained odor descriptors of the well known Dravnieks data set (see Fig.1a) \cite{dravnieks1985}. The ratings are obtained by asking human raters to assign values, on a fixed scale, of how close their perceptual experience of smelling an odorant is to each one of the descriptors; e.g. “on a scale from 0 to 5, how fruity do you perceive this smell?” (see Methods).

- Extending future work sections. It could be interesting to provide an overview of the work to be done in order to be able to fully predict natural language description of smells. Authors’ contributions should also be discussed according to it.

...

- Discussion related to the limits of the approach should be extended.

[We agree with the reviewer, and we added the following paragraph at the end of the Discussion section in response:]

Moreover, our findings are also clinically relevant, given that changes in olfactory perception are one of the first signatures of Alzheimer's Disease \cite{devanand2000} and associated with a range of other mental disorders \cite{corcoran2005}. Our approach provides a means to assess directly how these perceptual disturbances are associated with cognitive and emotional states.

Several limitations of the current approach need to be mentioned, along with possible ways to overcome them. In the first place, the model needs to be extended to mixtures of molecules; a naive linear superposition may suffice, but there is strong evidence that mixtures are particularly susceptible to non-linear interactions \cite{joerges1997}. Secondly, as already mentioned, it is possible to enlarge the basic distributional semantic model with additional lexical structure not easily captured by context-as-semantics hypothesis, such as synonyms/antonyms, part-of-speech markers such as verbs and nouns, so as to minimize the distortions we observed in the semantic-to-perception mapping. Related to this last issue, it remains to be seen how the word-based approach presented here will be extended to unconstrained discourse, in particular as it pertains to the expected difference between open narratives of the olfactory perceptual experience by smell experts and untrained raters \cite{medjkoune2016}. We hope that, for all these extensions, our work will provide a foundation to build upon.

- Provide a related work section: e.g. cite other contributions related to semantic analysis of odor (also using word embeddings), e.g. Medjkoune et al. 2016 Towards a Non-oriented Approach for the Evaluation of Odor Quality. Proceedings, Part I of the 16th International Conference, IPMU 2016, Eindhoven, The Netherlands, June 20-24, 2016, ISSN: 1865-0929. DOI:10.1007/978-3-319-40596-4_21

[We again agree with the reviewer that we didn't fully cover the literature, and now provide an explicit related work paragraph in the Introduction and added the suggested citation in the discussion:]

Recent work even suggests that olfactory knowledge can improve the performance of linguistic representations in predicting human similarity judgments \cite{kiela2016}, while linguistic representations can be applied to quantify the olfactory specificity and familiarity of words \cite{iatropoulos2018}. Other recent work uses the linguistic approach to predict a reduced representation, via clustering, of the odor of a molecule \cite{nozaki2018}; however, the predictive efficacy of this model falls abruptly when more than 5 clusters of descriptors are considered.

- Modify the title of the paper; it is not in accordance with the contribution – smells are most often more complex than mono-molecular odorants.

[We have modified the title of the paper to “Predicting natural-language descriptions of monomolecular odorants.”]

- The benefits and the impact of the contribution introduced in the conclusion (p8) should be presented in the introduction to motivate the study.

[In accordance with the reviewer's comment, we have modified the last paragraph in the introduction to read as follows:]

Such an approach eliminates the need for the time-consuming tasks of handcrafting domain-specific sets of olfactory descriptors and collecting ratings for large numbers of descriptors and odorants \cite{wise2000,kaepler2013,noble1984,lawless2013}, while establishing that the semantic distance between descriptors defines the equivalent of an odorwheel.

- The correspondence between perceptual and semantic spaces should be further studied. Differences between those spaces could have been illustrated to underline the difficulty of the task, e.g. by showing distortions (regions that differ strongly in term of distances). Additionally, considering provided title, advanced discussions related to the used of more complex linguistic units such as simple nominal groups are expected.

[This is an excellent point indeed, which we have ignored in the interest of brevity and simplicity of the message. While probably several papers can be written just on this topic, we have added a new figure, Figure 2-c&d, to illustrate the comparison between perceptual and semantic spaces, which as the reviewer suggested can be interpreted as an isomorphic map punctuated by distortions. We added in consequence a new section, “Correspondence between semantic space and olfactory ratings space,” where we added the following:]

Further insight is gained from looking at arrangement changes of two-dimensional projections of the DREAM descriptors based on their ratings distance (Fig.2c) and their semantic distance (Fig.2d; also see Methods). Notably, we observe only small local distortions of group mappings, e.g. “grass”, “flower” and “fruit” contiguous in both spaces (pink). However, there is also a global distortion as “sweet” is arranged in the semantic space near its antonym “sour,” and in the ratings space “sweet” is arranged closer to the perceptually similar term “bakery,” and “sour” is arranged closer to the perceptually similar term “decayed”.

[Additionally, we discuss the correspondence in subsection “Organization of descriptors in semantic and olfactory ratings spaces” as follows:]

On the other hand, the incomplete correspondence between the semantic and olfactory spaces is reflected in the failure to incorporate higher-order semantic concepts such as synonymy/antonymy, meronymy/hypernymy, etc., which could be leveraged to improve our model \cite{budanitsky2006} (Fig. 2c&d).

[And return in the Discussion section to this issue, to suggest future work aimed at overcoming the limitations inherent to the semantic embedding methodology we chose:]

Secondly, as already mentioned, it is possible to enlarge the basic distributional semantic model with additional lexical structure not easily captured by context-as-semantics hypothesis, such as synonyms/antonyms, part-of-speech markers (e.g. verbs, nouns), etc., so as to minimize the distortions we observed in the semantic-to-perception mapping.

- The source code used to perform the experiments should be publicly available in order to be able to reproduce the results.

[We will make the source code publicly available on Github before publication.]

- Performance of models should be discussed considering inter-agreement of experts.

[This is an excellent point. Knowing the inter-agreement of the experts who produced the Dravnieks ratings would set a ceiling on the performance of an ideal model for the prediction of

odors. Unfortunately, we do not have access to the individual ratings produced by the experts in the Dravnieks study. Future work might aim to collect expert ratings for this set and determine the inter-rater agreement ceiling.]

- It is not clear why the authors are using a correlation instead of a more classical quadratic error to evaluate the approaches.

[We used correlation to make our results commensurate with those presented in Keller et al. (2017). One advantage of using correlation is that it ranges between -1 and 1, so it is easy to interpret in absolute terms. Another is that it fits well with the idea that an odor's characterization is a sort of profile of the relative strengths of different odor notes or impressions, where the absolute strengths depend on concentration.]

- It could be interesting to comment the impact of the choice of the method – and (hyper)-parameters- used to build embeddings (word2vec, Glove, using various parameters such as different vector sizes). Word embeddings are the cornerstone of the contribution; such technical results are important in this context.

[Originally we used word2vec vectors trained on Google News. While performance was slightly improved by fastText, the difference was small and was not enough to affect the significance of the results. In response to the author's comments, we briefly mention this in the Methods section of the revised text.]

- More complex non-linear state-of-the-art models could have been used (non-linear SVM) and should be tested. The fact that only linear models have been tested should at least be motivated.

[With such a small sample size (in this case, each molecule is a sample), the ability to learn complex models from the data is limited. In the DREAM challenge where the goal was to predict the perceptual ratings of monomolecular odorants from their chemoinformatic properties (Keller et al. 2017), it was found that simple regularized linear regression outperformed or performed similarly to more complex methods. Nevertheless, we did attempt to use some more complex models during exploratory analysis of the data, focusing on models meant for efficient learning in cases of data scarcity. Namely, we looked at the semantic autoencoder for zero-shot learning (Kodirov et al., CVPR 2017), a non-linear neural-network model. We also looked at using partial least-squares regression, and 2-blocks canonical partial least-squares (Tenenhaus, 1998).]

- Please detail what is the similarity measure used to compare the embeddings (cosine similarity and Euclidian are mentioned, the one used in practice is not specified).

[We used the cosine similarity measure to compare the embeddings in the dendrogram. We now mention this in the “Organization of semantic and rating spaces” subsection of the Results section.]

- The protoDash algorithm should be detailed or removed.

[We agree that the introduction of ProtoDash was not properly motivated. We have modified and extended the motivation to use a prototype selection algorithm, and kept a detailed exposition of ProtoDash in Methods:]

The method we used for prioritizing the 19 perceptual descriptors is a state-of-the-art prototype selection algorithm based on a non-negative constrained reconstruction of the original data (see Methods and \cite{proto}). We chose this approach as it selects recursively the best \{sl individual\} descriptor, i.e. the descriptor that best explains the entire perceptual ratings data, as opposed to commonly-used dimensionality-reduction factorization algorithms.

Reviewer 2:

Recommendation:

Comments:

Title: Predicting natural language descriptions of smells

Summary: The manuscript described above shows the method to predict odor impression using natural language semantic representation. When chemoinformatic features such as molecular features for each molecule are given, the ratings of odor descriptors can be predicted. The authors mentioned the semantic-based approach achieved high accuracy to predict perceptual ratings of smell.

Although the level of the manuscript is relatively high, the acceptance of this manuscript as a paper of Nature Communications is not recommended because of its originality. The following paper using natural language semantic representation to predict impression of odor has been already published although the input data for prediction are different.

Predictive modeling for odor character of a chemical using machine learning combined with natural language processing, Yuji Nozaki and Takamichi Nakamoto, PLoS ONE 13(6): e0198475. <https://doi.org/10.1371/journal.pone.0198475>, June 14, 2018.

Since the main point of this work is to use natural language semantic representation to predict odor impression, its novelty is weak for Nature communication. It should represent an advance which is likely to influence thinking in the field.

However, I think that the manuscript includes several new things and ideas acceptable for other journal if the authors modify their claims. I suggest that the authors can reorganize the manuscript and submit it to other journal. The following comments might be helpful to reorganize the manuscript.

Figure 1

The reason for using two data sets such as Dream and Dravnieks ones are not clearly described. It seems better to use one unified data set. Since the authors used two data sets, the data flow became complicated. Moreover, it is not clear whether the accuracy to convert from Dream ratings to Dravnieks ratings is sufficient.

How many descriptors are necessary to express an odor impression? If 19 descriptors are enough, why 131-descriptor data set was used together? If 131 descriptor-approach is the best way, 19-descriptor data set lacks some information.

Figure 2 a

It is not clear how direct semantic method is used to predict Dravnieks rating. Transformation S maps semantic descriptor vector of dream onto the semantic descriptor vector of Dravnieks. It does not mean the transformation of dream rating vector into Dravnieks rating vector.

Figure 2 a bottom

Although the authors claim that the direct semantic model uses no molecule during training, why does the lateral axis mean number of Dream/Dravnieks overlap training molecules.

Figure 2 a,b bottom

In my understanding, the authors claim the advantage of direct semantic model. However, direct rating model is always better than direct semantic model when the number of training molecules was large. Thus, a user should use direct rating model together with large number of training molecules.

Figure 3 a

Although the correlation increases as the number of descriptors increases, the value of correlation seems low. The authors should comment on this point.

Figure 3 b, c

The authors might claim that mixed model is better than direct semantic model. However, the meaning of mixed model is the same as that of direct rating model since it uses molecule information. The authors should comment on this point.

In abstract (Page 2), the authors mentioned that establishing that the semantic distance between descriptors defines the equivalent of an odorwheel.

In Figure 3 d, there are many descriptors with low correlation across molecules. Can the authors say that they have established the method? Otherwise did they just find the possibility? The author should describe the level of completion.

Figure 4

Why not all but 80 descriptors were used here? The authors mentioned 7 overlap descriptors. What groups of descriptors share 7 descriptors?

Figure 4 b

Where are qualitative sensory description coming from? Is it possible to show its rating in comparison with true values?

Response to Reviewer 2 (general): [We thank the reviewer for the numerous suggestions to improve the readability of the manuscript, which we address below. We disagree however with

the reviewer's characterization of unoriginality of our results based on prior work on linguistic representations for the prediction of odor qualia. We don't claim this has never been done before, much less that it has been considered - note that we refer to extensive literature of the influence of language on perception in the second paragraph of the Introduction. Our claim is of an efficient predictive generalizable model, i.e. a comprehensive solution for predicting arbitrary descriptors over arbitrary mono-molecular odors. We don't claim the problem: we claim the solution. We hope we can convince the reviewer this solution is original.]

Response to each comment of Reviewer 2 (individually):

Although the level of the manuscript is relatively high, the acceptance of this manuscript as a paper of Nature Communications is not recommended because of its originality. The following paper using natural language semantic representation to predict impression of odor has been already published although the input data for prediction are different.

Predictive modeling for odor character of a chemical using machine learning combined with natural language processing, Yuji Nozaki and Takamichi Nakamoto, PLoS ONE 13(6): e0198475. [https:// doi.org/10.1371/journal.pone.0198475](https://doi.org/10.1371/journal.pone.0198475), June 14, 2018.

Since the main point of this work is to use natural language semantic representation to predict odor impression, its novelty is weak for Nature communication. It should represent an advance which is likely to influence thinking in the field.

However, I think that the manuscript includes several new things and ideas acceptable for other journal if the authors modify their claims. I suggest that the authors can reorganize the manuscript and submit it to other journal. The following comments might be helpful to reorganize the manuscript.

[We thank the reviewer for pointing out this recently published paper, which we have added as a reference in our manuscript. However, for several reasons, we disagree that this publication detracts from the originality or general impact of our study. First of all the article indicated by this reviewer was made public after our manuscript was made available through biorxiv on May 25, 2018 and submitted for review. Second, we never claimed that the originality or interest of our manuscript was solely based on the use of semantic embeddings. In fact, in reference 7 of our manuscript we cite another paper where semantic embeddings are applied to olfaction research, **Kiela et al (2015)**. Third and most importantly, although both studies apply semantic embedding to olfaction research, our contribution is to show that this approach can be effectively used to predict perceptual ratings over each one of a large and potentially arbitrary set of descriptors. **Nozaki & Takamichi** and **Kiela et al** are not able to accurately predict ratings of single descriptors, but have to aggregate descriptors into clusters to predict whether they are being activated or not and the predictive efficacy of this model falls abruptly when more than 5 clusters of descriptors are considered. Also these studies neither predict rating values for an arbitrary set of single molecules, nor distinguish domain-specific sets of descriptors or odor wheels.]

The reason for using two data sets such as Dream and Dravnieks ones are not clearly described. It seems better to use one unified data set. Since the authors used two data sets, the data flow became complicated. Moreover, it is not clear whether the accuracy to convert from Dream ratings to Dravnieks ratings is sufficient.

[We thank the reviewer for his comment and we modified the manuscript in order to simplify the flow and clarify the main points of our study; see the detailed reply to Reviewer 1 in this respect, as his/her comments are very much aligned. The DREAM dataset was essential as this helped us train our models to predict rating values for a smaller set of descriptors and then expanding them to the Dravnieks descriptors. This also allowed the prediction of ratings for arbitrary sets of molecules.]

How many descriptors are necessary to express an odor impression? If 19 descriptors are enough, why 131-descriptor data set was used together? If 131 descriptor-approach is the best way, 19-descriptor data set lacks some information.

[We thank the reviewer for this interesting comment. The number of descriptors necessary to obtain an adequate odor perception is not an absolute parameter and depends on the odor domain, the subjects and the purpose. In this study we have found that the 19 semantic descriptors of DREAM as sufficient to predict with half of the 131 Dravnieks descriptors with a correlation of 0.5 or higher.]

It is not clear how direct semantic method is used to predict Dravnieks rating. Transformation S maps semantic descriptor vector of dream onto the semantic descriptor vector of Dravnieks. It does not mean the transformation of dream rating vector into Dravnieks rating vector.

[The same matrix S used to transform semantic descriptors was used to transform ratings. We clarified this point in the first paragraph of the “Extending predictions to arbitrary descriptors” subsection of the Results section.]

The method we used for prioritizing the 19 perceptual descriptors is a state-of-the-art prototype selection algorithm based on a non-negative constrained reconstruction of the original data (see Methods proto)). We chose this approach as it selects recursively the best individual descriptor, i.e. the descriptor that best explains the entire perceptual ratings data, as opposed to commonly-used dimensionality-reduction factorization algorithms.

Although the authors claim that the direct semantic model uses no molecule during training, why does the lateral axis mean number of Dream/Dravnieks overlap training molecules.

[As indicated above, the S matrix used to transform semantic vectors is also used to transform rating vectors and hence no molecular data was used for the direct semantic model using only S . The lateral axis relates mainly to the ratings and mixed models that do use molecular ratings. The direct semantic model, like the other models, is affected by the fact that the background model values – i.e for each descriptor the average value across all training molecules– changes according to the molecular ratings used. We hope this clarifies the issue.]

In my understanding, the authors claim the advantage of direct semantic model. However, direct

rating model is always better than direct semantic model when the number of training molecules was large. Thus, a user should use direct rating model together with large number of training molecules.

[This is precisely our point, as in most situations a large amount of training molecules for predictions is not available and obtaining it is expensive, using the direct semantic model in situations of data scarcity turns out to be a very good prediction and the best option available. Also note that the number of experimental data points required to learn a ratings model is expected to be proportional to the dimensionality of the odor descriptor set, making an experimentally-based comprehensive solution to the stated problem even more challenging.]

Although the correlation increases as the number of descriptors increases, the value of correlation seems low. The authors should comment on this point.

[The observation of the reviewer is accurate, the correlations for the predictions are low and this is due to several reasons: 1) No molecular data is used for the predictions of the direct model 2) In the imputed model not even ratings from DREAM are used 3) A smaller amount of information is used as less information from descriptors are included. As the reviewer can see, the correlation grows with the descriptors included. Once all the descriptors are included for training the correlations reach the same level as in the inset of Figure 2b.]

The authors might claim that mixed model is better than direct semantic model. However, the meaning of mixed model is the same as that of direct rating model since it uses molecule information. The authors should comment on this point.

[As discussed above, the direct semantic model using only **S** trained on semantic vectors uses no molecular data. Hence the mixed model is indeed different than the direct model as its predictions are built calculating the average of the direct semantic and the ratings model. There are furthermore deeper conceptual reasons to understand the difference between the models, regardless of the specific results. The mixed model can be understood as combining direct perceptual ratings of a handful of participants with extensive perceptual judgments expressed in language use by the population at large. In this sense, we can interpret the mixed model as containing additional experimental data.]

In abstract (Page 2), the authors mentioned that establishing that the semantic distance between descriptors defines the equivalent of an odorwheel.

In Figure 3 d, there are many descriptors with low correlation across molecules. Can the authors say that they have established the method? Otherwise did they just find the possibility? The author should describe the level of completion.

[The reviewer's observation is indeed accurate, only half of the 131 descriptors are predicted with a correlation higher than 0.5, but this is 10-fold better than any previous attempt (Keller et al 2017). In the future directions paragraph of the Discussion section we indicate several options for trying to obtain even better predictions for all descriptors.]

Why not all but 80 descriptors were used here? The authors mentioned 7 overlap descriptors. What groups of descriptors share 7 descriptors?

[In figure 4 the set of 80 descriptors is a completely new one, obtained from: <http://www.thegoodscentcompany.com/> and used to describe the families of molecules whose odor qualities we tried to predict.

As indicated, the 7 overlap descriptors between the 19 DREAM descriptors and the 80 in this new compilation are acid, floral, fruity, sour, sweaty, sweet and wood. We hope that adding this in the manuscript will satisfy the reviewer.]

Where are qualitative sensory description coming from? Is it possible to show its rating in comparison with true values?

[As discussed above, the qualitative sensory description for these molecular families was obtained and compiled from: <http://www.thegoodscentcompany.com/>
Unfortunately no ratings values are present and we could not obtain them from any other source.]

Reviewer 3:

Recommendation:

Comments:

This is a rigorous and interesting application of machine learning to an important question about odor perception. Congrats to the authors on a novel and nicely executed study.

Describing smells with language is notoriously difficult and open ended, and begs the question of how much our verbal labels for odors are anchored by olfactory experience vs. ‘simply’ reflective of the much broader semantic life and relationships that words have among themselves. In other words, if someone had a crummy sense of smell, but a great vocabulary, could they still describe odors using all the “right” words? This study says yes, and that is an important result.

I don’t have any major technical concerns and found the work to be thorough. I think it would be an important addition to the literature that would arouse broad interest.

Major comments:

1) I’m not sure I arrive at the same conclusion from the comparison between Fig. 1b and 1c that the authors do. Namely, that the two correlation matrices are similarly structured, and given that, “favor the hypothesis of a tight perceptual-linguistic bond between the descriptors ratings and their linguistic meanings.” If that were the case, shouldn’t we see some of the same block structure in 1c as in 1b? Instead, 1c seems rather sparse, with only a few strong, singular hits showing up in those places where the semantic descriptor vectors will be identical. I know the significance tests say the matrices are similar, but I wonder if that could be driven by a relatively small subset descriptors for which the “tight link” hypothesis holds very strongly, while the hypothesis is much less compelling across the whole of descriptor space. (More below).

2) Related: from 1c, the only case for which I can convince myself that there's some broad correspondence between DREAM semantic descriptors and Dravnieks descriptors is for the {Flower, Sweet, Fruit} block that the authors emphasize. One could imagine that there are a subset of words that, when used, tend to be used in contexts where one is describing odors. Many other words, however, might have their meaning distributed across many additional contexts, and are only occasionally co-opted to describe odors. Did you/could you look to see whether dropping certain specific descriptors (or combinations) has much more dramatic consequences for your models than others? It would be useful to know if dropping {'Flower' and 'Sweet'}, for example, has consequences that were little distinguishable from dropping any other pair descriptors.

Minor comments:

The 'DREAM ratings' and 'Dravnieks ratings' schematics you show in figure 2 are confusing, and I'm not sure they do much to clarify what you're doing. I had to zoom in 250% before I could really see that they were supposed to represent ratings sliders. I'd recommend trying to come up with a more compelling visual.

P14, legend: "...and they have 58 molecules are common to both data sets." → "... they have 58 molecules in common to both..."

Response to Reviewer 3 (general): [We thank very much the praise expressed by the reviewer, and in particular his/her acknowledgment of the difficulty of describing olfactory perception with language. We also took equally serious the comments and suggestions to improve the manuscript, addressed below.]

Response to each comment of Reviewer 3 (individually):

1) I'm not sure I arrive at the same conclusion from the comparison between Fig. 1b and 1c that the authors do. Namely, that the two correlation matrices are similarly structured, and given that, "favor the hypothesis of a tight perceptual-linguistic bond between the descriptors ratings and their linguistic meanings." If that were the case, shouldn't we see some of the same block structure in 1c as in 1b? Instead, 1c seems rather sparse, with only a few strong, singular hits showing up in those places where the semantic descriptor vectors will be identical. I know the significance tests say the matrices are similar, but I wonder if that could be driven by a relatively small subset descriptors for which the "tight link" hypothesis holds very strongly, while the hypothesis is much less compelling across the whole of descriptor space. (More below).

[We agree that Figure 1c, as presented, seems to show very few non-zero hits. However, this is an artifact of the display choices made in producing the figure. In fact, rather than being sparse, nearly all of the correlations/cosines are nonzero, because all of the vectors lie in a common subspace of the semantic space (due to all being sensory words and thus co-occurring frequently). To illustrate more clearly the correspondence between the spaces we completed the Matrices with a different representation in Figure 2c&d. Please see the reply to Reviewer 1 for the specific changes introduced in the manuscript, given that his/her comments we similar in this respect; note in particular how the new figure illustrates the limitations of the current semantic embedding approach, and at the same time suggests ways to overcome them.]

2) Related: from 1c, the only case for which I can convince myself that there's some broad correspondence between DREAM semantic descriptors and Dravnieks descriptors is for the {Flower, Sweet, Fruit} block that the authors emphasize. One could imagine that there are a subset of words that, when used, tend to be used in contexts where one is describing odors. Many other words, however, might have their meaning distributed across many additional contexts, and are only occasionally co-opted to describe odors. Did you/could you look to see whether dropping certain specific descriptors (or combinations) has much more dramatic consequences for your models than others? It would be useful to know if dropping {'Flower' and 'Sweet'}, for example, has consequences that were little distinguishable from dropping any other pair descriptors.

[We did not directly explore the impact of including or omitting individual descriptors based on their odor specificity, we did look at whether odor specificity of a descriptor (as measured using the method of Iatropoulos et al. (2018)) was associated with the prediction performance impact of omitting that descriptor. We did not find a significant pattern when we performed these analyses. Let us note however that Figure 4A does provides a partial answer to this]

The 'DREAM ratings' and 'Dravnieks ratings' schematics you show in figure 2 are confusing, and I'm not sure they do much to clarify what you're doing. I had to zoom in 250% before I could really see that they were supposed to represent ratings sliders. I'd recommend trying to come up with a more compelling visual.

[We completely agree and thank the reviewer for pointing this out, and we have changed the arrangement of our figures and the visuals used in order to make things more intuitive.]

REVIEWERS' COMMENTS:

Reviewer #1 (Remarks to the Author):

The authors have made a substantial work to improve the paper based on reviewers' remarks. In particular, regarding the remarks I proposed (reviewer 1), I consider that the authors have provided enough detailed answers and corresponding modifications to answer them (additional details, figures and results, as well as a commitment to publish the source code). I therefore consider that the proposed work addresses the limitations underlined in my previous review.

Reviewer #3 (Remarks to the Author):

Congrats again to the authors on a nicely executed and important study. My critiques (which were modest to begin with) have been addressed satisfactorily, and I recommend that the manuscript be considered for publication.